# Glutamatergic cerebellar neurons differentially contribute to the acquisition of motor and social behaviors

Meike E. van der Heijden [1,2,5], Alejandro G. Rey Hipolito[2,3,5], Linda H. Kim [1,2], Dominic J. Kizek [1,2], Ross M. Perez[2,4], Tao Lin [1,2] & Roy V. Sillitoe [1,2,3,4] ✉

Insults to the developing cerebellum can cause motor, language, and social deficits. Here, we investigate whether developmental insults to different cerebellar neurons constrain the ability to acquire cerebellar-dependent behaviors. We perturb cerebellar cortical or nuclei neuron function by eliminating glutamatergic neurotransmission during development, and then we measure motor and social behaviors in early postnatal and adult mice. Altering cortical and nuclei neurons impacts postnatal motor control and social vocalizations. Normalizing neurotransmission in cortical neurons but not nuclei neurons restores social behaviors while the motor deficits remain impaired in adults. In contrast, manipulating only a subset of nuclei neurons leaves social behaviors intact but leads to early motor deficits that are restored by adulthood. Our data uncover that glutamatergic neurotransmission from cerebellar cortical and nuclei neurons differentially control the acquisition of motor and social behaviors, and that the brain can compensate for some but not all perturbations to the developing cerebellum.

Cerebellar injury in preterm infants is often associated with movement disorders, language impairments, and social deficits[1,2]. Preterm injuries affect the exponential phase of granule cell proliferation, although they often also alter the early development of all glutamatergic neurons in the cerebellum. The resulting defects lead to long-term changes in gray matter volume in the cerebellar cortex that are correlated to the severity of neural deficits in infants[3–6]. Cerebellar cortical injuries further impact the development and function of downstream cerebellar nuclei neurons, which serve as the predominant output from the cerebellum and link it to the rest of the motor network[7]. Accordingly, cerebellar defects can also impair the development and function of the cerebral cortex[8,9]. The combined injury to the cerebellum and neocortex may help explain the broad neural deficits observed in many infants that experience cerebellar injury during the perinatal period.

Accumulating evidence suggests that the site of injury within the developing cerebellum may determine behavioral outcomes. When damage is confined to the cerebellar nuclei neuron axons that project and travel through the superior cerebellar peduncle, affected patients (typically children) can develop posterior fossa syndrome, which is hallmarked by ataxia, mutism, and changes in social interactions[10,11]. Intriguingly, over time, patients largely recover these impaired neural functions with only minor residual symptoms, most commonly persisting in their motor coordination[12,13]. Patients and apes with lesions localized to the cerebellar nuclei mainly demonstrate motor symptoms with limited recovery following the injury[14,15]. When the cerebellar cortex is the primary site of the lesion, deficits in social cognition, language, and motor performance arise, but they often persist following the initial injury with limited recovery, especially in the non-motor domains affecting cognition, sociability, language, and affect[1,2].

[1]Department of Pathology & Immunology, Baylor College of Medicine, Houston, TX, USA. [2]Jan and Dan Duncan Neurological Research Institute at Texas Children's Hospital, Houston, TX, USA. [3]Department of Neuroscience, Baylor College of Medicine, Houston, TX, USA. [4]Development, Disease Models & Therapeutics Graduate Program, Baylor College of Medicine, Houston, TX, USA. [5]These authors contributed equally: Meike E. van der Heijden, Alejandro G. Rey Hipolito. ✉e-mail: sillitoe@bcm.edu

Similarly, studies in rodents have provided compelling evidence that disrupting cerebellar cortical function during development can lead to motor impairments, altered vocalizations, and social deficits[16–21].

These clinical outcomes illustrate that while damage to the cerebellar cortex or to the downstream cerebellar nuclei neurons during infancy is sufficient to impair motor function, language, and social behavior, there are instances when the cerebellum can remarkably overcome perturbations and restore functions. Importantly, the degree of compensation may be linked to the cerebellar neurons that are primarily affected by the lesion, suggesting a unique role for each cerebellar neural subtype in the regulation of cerebellar-associated behaviors. These studies have inspired the need for a deeper examination of how the relatively few neuron types in the cerebellum contribute to a wide range of motor and non-motor functions. However, it remains largely unknown where in the circuit cerebellar-associated behaviors originate, whether the same neuronal subtypes contribute equally to the acquisition of these diverse behaviors, and whether the perturbation of all neuronal subtypes can be equally compensated for during development. Additionally, it remains specifically unexplored how the cerebellar nuclei contribute to the acquisition of different behaviors during postnatal life.

Dissecting how cerebellar cortical and cerebellar nuclei neurons contribute to the acquisition of different cerebellar-dependent behaviors requires the use of non-invasive and cell-type specific manipulations during circuit development. Fortunately, the cellular architecture of the cerebellar anlage lends itself to precise genetic approaches. The embryonic cerebellum arises from two distinct lineages that interact to form the cerebellar circuits[22]. The unique identities of the lineages can be used to manipulate GABAergic or glutamatergic cerebellar neurons. The *Ptf1a* lineage that originates in the ventricular zone gives rise to GABAergic neurons, including GABAergic cerebellar nuclei neurons and Purkinje cells[23]. In contrast, the *Atoh1* lineage is derived from the rhombic lip and gives rise to glutamatergic neurons, including glutamatergic granule cells and cerebellar nuclei neurons (Fig. 1a, b, c)[24,25]. Granule cells are the most abundant neuron type in the cerebellum, and they provide the predominant synaptic input to Purkinje cells[26]. In turn, Purkinje cells send convergent projections to their principal targets–glutamatergic and GABAergic cerebellar nuclei neurons[27]. The cerebellar nuclei neurons form the main output of the cerebellum through parallel pathways that project throughout the brain and spinal cord[28].

We previously showed that we could take advantage of the embryonic *Atoh1* lineage to specifically manipulate the neurogenesis of glutamatergic cerebellar neurons[29]. In the central nervous system, *Atoh1* functions as a pro-neural gene that is necessary for the neurogenesis of glutamatergic neurons; these neurons are born along the most dorsal portion of the developing brainstem and spinal cord[30,31]. The contribution of *Atoh1* lineage neurons to behavior has relied mainly on the use of conditional knockout mice because *Atoh1* lineage neurons are essential for respiration, and as a consequence, *Atoh1* null mice die at birth[24,32]. We recently showed that preventing the neurogenesis of glutamatergic, *Atoh1* lineage granule cells and cerebellar nuclei neurons impairs the early postnatal acquisition of cerebellar-dependent behaviors, including motor coordination and vocalizations in a social isolation paradigm[29]. A primary finding in our study was that the neurogenesis of granule cells was also essential for the maturation of Purkinje cell activity. However, by using an approach that eliminated neurogenesis, we could not delineate whether the resulting behavioral deficits were due to granule cell loss, abnormal Purkinje cell signaling, glutamatergic cerebellar nuclei neuron loss, or a combination thereof.

To investigate whether the motor impairments and social deficits in *Atoh1* conditional knockout mice are due to the loss of functional output from glutamatergic cerebellar neurons, we set out to examine *Atoh1* lineage neuron function using a gene silencing approach. Specifically, we deleted the gene encoding a vesicular transporter for glutamate (*Vglut2*) to eliminate the transport of glutamate into the synaptic vesicles of *Atoh1* lineage neurons. This manipulation results in an effective silencing of fast neurotransmission in the genetically targeted neurons (Fig. 1a)[33]. Then, to better define how VGluT2-mediated neurotransmission from the nuclei neurons contributes to cerebellar-dependent behaviors, we deleted *Vglut2* from *Ntsr1*-expressing cells, an approach that predominantly targets the glutamatergic nuclei neurons[34]. In addition, we leveraged an interesting developmental transition from VGluT2- to VGluT1-mediated neurotransmission that uniquely occurs in granule cells to investigate whether restoration of cerebellar cortical function improves neural deficits (Fig. 1a, b, c). This allowed us to assess whether the cerebellar-dependent motor and social deficits are improved following the natural restoration of granule cell neurotransmission in the developing cerebellar circuit. Finally, we used *Ntsr1Cre;Vglut2fl/fl* conditional knockout mice to investigate whether neural functions are restored by developmental compensation regardless of changes in vesicular subtype switching. Together, this combination of genetic approaches provides us with in vivo cell-type specific methods for precisely lesioning neural function during development. Using this strategy, we investigated how different cerebellar neural subtypes contribute to the acquisition of diverse cerebellar-dependent behaviors and demonstrated the exceptional ability of the cerebellum to regain neural functions after developmental perturbations.

## Results

### Conditional *Vglut2* deletion from the *Atoh1* lineage affects early postnatal granule cells and glutamatergic cerebellar nuclei neurons throughout life

We selectively deleted the vesicular transporter for glutamate (*Vglut2*) from the *Atoh1* lineage, which resulted in a lack of VGluT2 protein in pre-synaptic vesicles of the *Atoh1* lineage, *Vglut2*-expressing neurons[33]. As a result, when an action potential arrives at the synapse, synaptic vesicles fuse to the pre-synaptic membrane but do not functionally affect the postsynaptic cells because no neurotransmitter is released into the synaptic cleft (Fig. 1a). In the current genetic manipulation, glutamatergic cerebellar nuclei neurons are affected throughout life (Fig. 1b, c). In contrast, granule cells express *Vglut2* only until the second postnatal week in mice[35], whereafter they switch their glutamate transporter to the type one transporter (*Vglut1*), and therefore the effects of our genetic manipulation do not actively and directly continue through adulthood. Unlike *Atoh1* null and cerebellum-specific *Atoh1* conditional knockout mice[24,29,32], *Atoh1Cre/+;Vglut2fl/fl* mice were viable and survived into adulthood, likely because *Atoh1Cre/+;Vglut2fl/fl* mice had normal respiratory rhythms (Supplementary Fig. 1).

Indeed, in situ hybridization (ISH) assays in adult animals showed that nearly all cerebellar nuclei neurons from the *Atoh1* lineage express *Vglut2* in adulthood (in *Atoh1Cre/+;Rosalsl-tdTomato/+* mice 95 ± 0.8% tdTomato+ neurons are also *Vglut2*+, N = 3 mice, n = 18 sections), but no *Vglut2* expression was found in the granule cell layer (Fig. 1d). We validated that our genetic manipulation considerably reduced *Vglut2* expression as *Vglut2* mRNA intensity was significantly reduced in *Atoh1Cre/+;Vglut2fl/fl* conditional knockout mice compared to control mice as measured by an ISH assay (normalized signal intensity; Control: 1 ± 0.08, N = 3, n = 21; *Atoh1Cre/+;Vglut2fl/fl*: 0.24 ± 0.04, N = 3, n = 16; LMM: p < 0.001) (Fig. 1e). Immunohistochemical staining for the VGluT2 protein confirmed that the genetic manipulation selectively reduced the proportion of VGluT2+ synapses in the molecular layer of P7 *Atoh1Cre/+;Vglut2fl/fl* mice throughout the cerebellar cortex (Fig. 1f). We visualized VGluT2+ synapses in crus I and lobule V because these cerebellar regions have been associated with social and motor functions, respectively[19,36]. In contrast, we did not detect differences in the quantity and distribution of VGluT2+ synapses between adult control and conditional knockout mice (Fig. 1g) since, at this age, only the

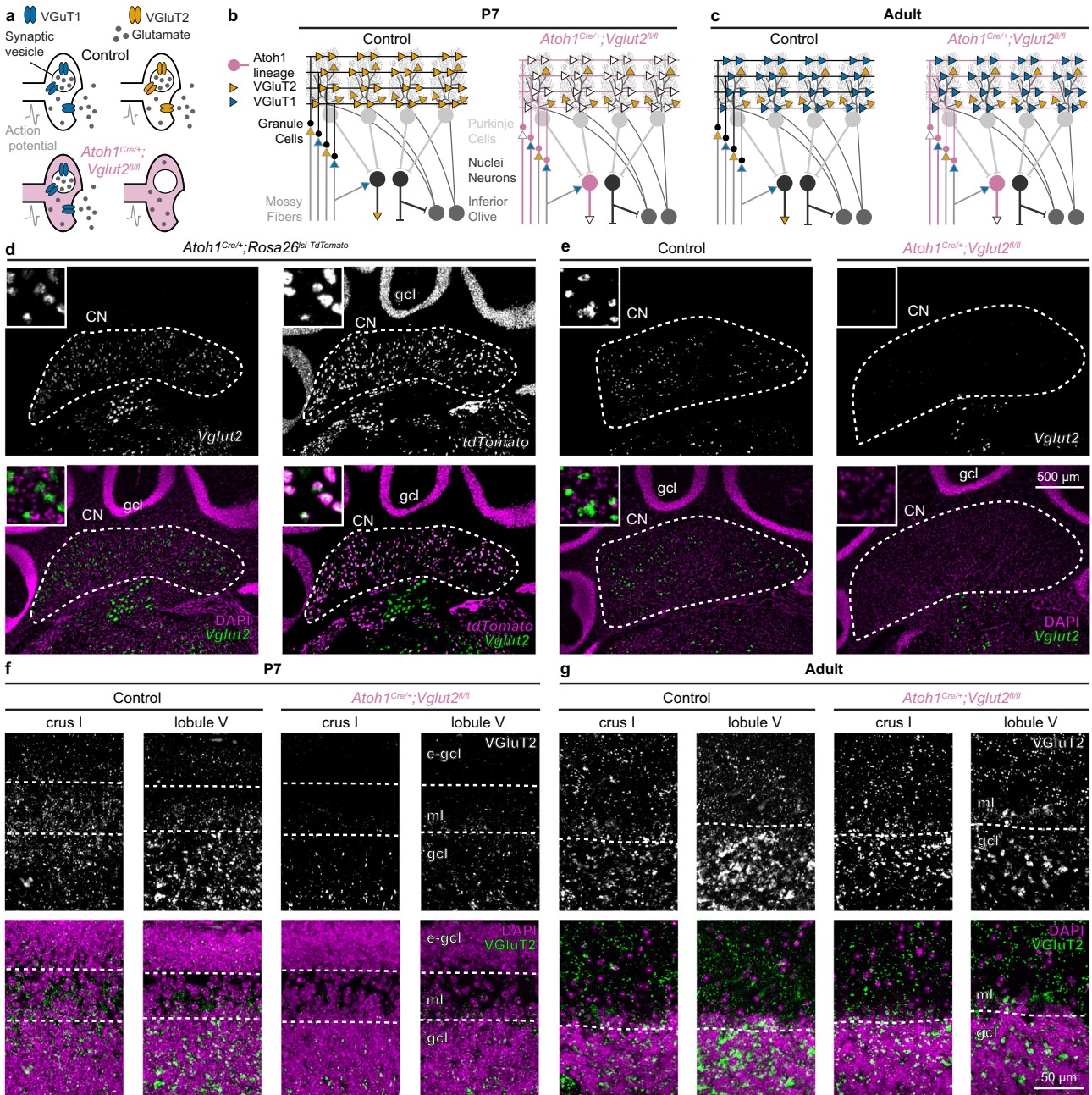

**Fig. 1 | Conditional *Vglut2* deletion from *Atoh1* lineage neurons. a** Schematic showing how conditional deletion of VGluT2 only affects fast neurotransmission in VGluT2-expressing neurons. **b** Schematic of cerebellar connectivity and vesicular transporter expression for the glutamate subtypes (VGluT1 and VGluT2) in the cerebellar circuit in P7 control mice and *Atoh1^Cre/+^;Vglut2^fl/fl^* conditional knockout mice. **c** Same as **b**, but in adult mice. For **a**–**c** VGluT1 (blue); VGluT2 (orange); Control mice (black); *Atoh1^Cre/+^;Vglut2^fl/fl^* mice (reddish purple). **d** Expression of *Vglut2* (green) with DAPI (purple, left) or *tdTomato* (purple, right) in the cerebellar nuclei (CN) and granule cell layer (gcl) of adult *Atoh1^Cre/+^;Rosa26^lsl-tdTomato^* mice. **e** Expression of *Vglut2* (green) and DAPI (purple) in the CN and gcl of adult control mice and *Atoh1^Cre/+^;Vglut2^fl/fl^* mice. **d**, **e** insets are 125 by 125 μm high magnification images. **f** VGluT2^+^ synapses (green) and DAPI (purple) in the molecular layer (ml) and gcl of P7 control and *Atoh1^Cre/+^;Vglut2^fl/fl^* mice, e-gcl = external granule cell layer containing actively proliferating granule cell precursor cells. **g** VGluT2^+^ synapses (green) and DAPI (purple) in the ml and gcl of adult control mice and *Atoh1^Cre/+^;Vglut2^fl/fl^* mice. **d**–**e** are shown at the same scale. **f**, **g** are shown at the same scale. Images are representative of *N* = 3 mice.

climbing fibers express VGluT2 in the molecular layer[21,35]. The lower density of VGluT2^+^ synapses in the granule cell layer of crus I relative to lobule V is in line with previous observations of differential innervation of spinocerebellar mossy fibers across the cerebellar cortex[37,38]. These results demonstrated the specificity of the conditional *Vglut2* deletion from the *Atoh1* lineage in the cerebellum and established the temporal dependence of VGluT2 in the *Atoh1* lineage cerebellar neurons.

Additionally, we found that *Atoh1^Cre/+^;Vglut2^fl/fl^* conditional knockout mice have no major differences in zonal stripe patterning of

VGluT2^+^ mossy fiber synapses in the anterior or posterior zones of the cerebellum, regions into which the spinocerebellar projections are targeted (Supplementary Fig. 2)[37–39]. This is in line with previous reports that *Atoh1* lineage neurons in the spinal cord are not the predominant source of spinocerebellar projections[40], and further suggests that loss of neurotransmission from granule cells and nuclei neurons does not affect overall zonal stripe formation in our conditional knockout mice. Finally, we did not observe changes in cerebellar morphology (Supplementary Fig. 3) or interneuron localization

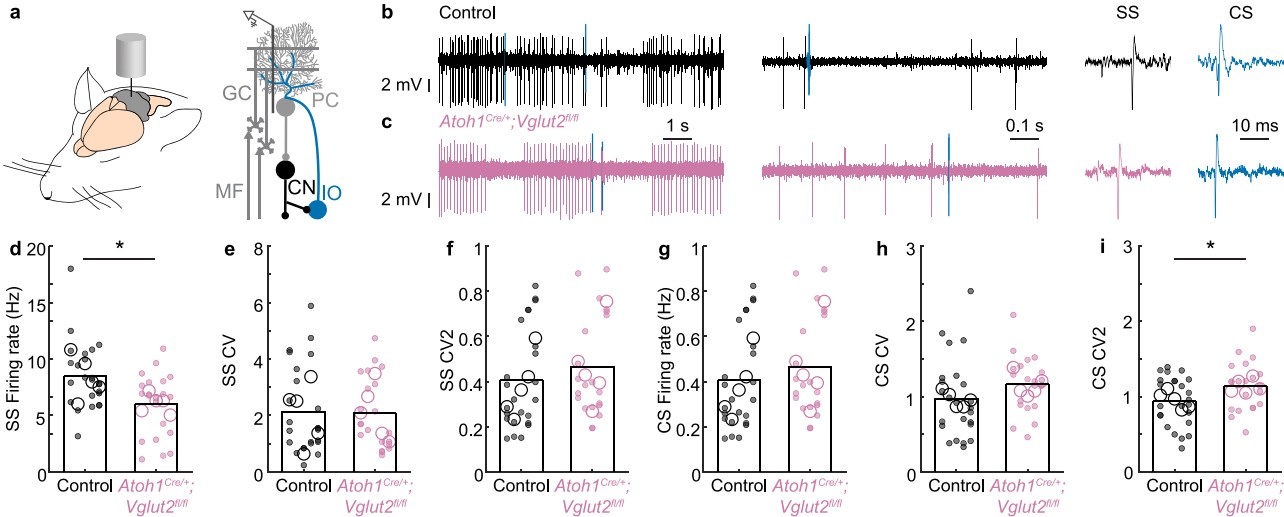

**Fig. 2 | Purkinje cell firing activity is abnormal in P10 *Atoh1*[Cre/+]*;Vglut2*[fl/fl] conditional knockout mice. a** Schematic of an in vivo single-unit extracellular Purkinje cell recording in an anaesthetized mouse. **b** Representative trace of a Purkinje cell recording in a control mouse (black and blue lines represent complex spikes). **c** Representative trace of a Purkinje cell recording in an *Atoh1*[Cre/+]*;Vglut2*[fl/fl] mouse (reddish purple, blue lines represent complex spikes). In **b**, **c** Inferior olive (IO) evoked complex spike (CS) in blue. The *y*-axis is constant across the panel. The *x*-axis (timescales) are the same for (**b**, **c**). **d** The frequency of Simple Spikes (SS) is different between control and *Atoh1*[Cre/+]*;Vglut2*[fl/fl] mice ($p = 0.005$; $d = 0.81$). **e** No differences were found in SS CV (spike pattern, $p = 0.977$; $d = 0.04$), **f** SS CV2 (spike regularity, $p = 0.407$; $d = 0.26$), **g** CS Frequency ($p = 0.548$, $d = 0.22$), or **h** CS CV

(spike pattern, $p = 0.171$, $d = 0.40$). **i** Complex spikes occurred more regularly in control than in *Atoh1*[Cre/+]*;Vglut2*[fl/fl] mice CS CV2 (spike regularity, $p = 0.039$, $d = 0.60$). For **d–i**, large open circles represent the mouse average; small, closed circles represent the cell average; data points from control mice in black, data points from *Atoh1*[Cre/+]*;Vglut2*[fl/fl] mice in reddish purple. A linear mixed model analysis with genotype as a fixed variable and mouse number as a random variable was used to test for statistical significance in (**d–i**). Control: $N_{Mice} = 5$, $n_{Cells} = 24$ cells; *Atoh1*[Cre/+]*;Vglut2*[fl/fl]: $N_{mice} = 5$, $n_{Cells} = 21$. Source data and detailed statistical results are available and provided as a Source Data file. Panel (**a**) was adapted from White & Sillitoe, 2017, "Genetic silencing of olivocerebellar synapses causes dystonia-like behavior in mice," Nature Communications[21] under CC BY 4.0.

(Supplementary Fig. 4) in our *Atoh1*[Cre/+]*;Vglut2*[fl/fl] conditional knockout mice, confirming that our genetic manipulation is restricted to the elimination of *Vglut2* expression in developing granule cells and cerebellar nuclei neurons throughout life.

### Purkinje cells in early postnatal *Atoh1*[Cre/+]*;Vglut2*[fl/fl] mice have abnormal firing activity

Next, we sought to confirm that the deletion of *Vglut2* from the *Atoh1* lineage neurons caused functional deficits in the cerebellar cortex during the early postnatal period. To achieve this, we performed in vivo single-unit recordings from Purkinje cells, which form the predominant downstream target of granule cells and are the sole output of the cerebellar cortex[22] (Fig. 2a). We randomly sampled spiking activity from a wide range of anatomical areas during our recordings, including but not limited to crus I and lobules III/IV/V (Supplementary Fig. 5). Purkinje cell firing patterns progress through a considerable series of maturation events during the second postnatal week in mice, in a process that is dependent on the neurogenesis of *Atoh1* lineage neurons[29]. During the in vivo recordings, Purkinje cells were identified by their unique spiking activity that consists of two distinct action potential types (Fig. 2b, c). Purkinje cells can generate simple spikes without synaptic input, and in the intact cerebellar circuit, simple spike firing frequency and patterns are modulated by inputs from local cerebellar cortical GABAergic interneurons and granule cells[41,42]. In contrast, complex spikes are mediated by a strong glutamatergic input from climbing fibers that originate in the inferior olive and can be recognized by their large initial waveform that is followed by ~3–5 smaller amplitude spikelets[43,44].

We postulated that the *Vglut2* deletion from the *Atoh1* lineage granule cells would reduce simple spike activity. We further investigated whether *Vglut2* elimination from granule cells would influence Purkinje cell spike pattern (CV) and regularity (CV2). In line with our hypothesis, we found that loss of *Vglut2* from the *Atoh1* lineage resulted in a reduction of simple spike firing rate (Fig. 2d) but no

change in simple spike pattern or regularity (Fig. 2e, f). We did not observe any differences in complex spike firing rate or pattern (Fig. 2g, i), but there was a slight increase in the irregularity of complex spikes (Fig. 2h, i), which may occur through secondary effects resulting from the loss of granule cell signaling and their influence on climbing fiber maturation[29,45,46]. Together, these results confirm that loss of neurotransmission in granule cells changes cerebellar cortical function.

### Postnatal developing *Atoh1*[Cre/+]*;Vglut2*[fl/fl] mice have disrupted motor function and impaired social vocalizations

After validating that our genetic approach was specific in its ability to target developing synapses, we next set out to examine whether genetically silencing *Vglut2*-expressing, *Atoh1* lineage neurons during development impairs the acquisition of cerebellar-dependent behaviors in the early neonatal period (Fig. 3a). We investigated the acquisition of motor functions and social behaviors that were also impaired upon blocking the neurogenesis of glutamatergic *Atoh1* lineage neurons[29]. *Atoh1*[Cre/+]*;Vglut2*[fl/fl] mice in the home cage often displayed abnormal dystonic movements and postures. To test how motor performance was affected in these mice, we tested two postnatally acquired motor reflexes that we previously showed are severely impaired in mice with postnatal dystonia[47]. First, we measured the negative geotaxis reflex by placing the mice head down on a negative slope and timing the latency for them to rotate upward (Fig. 3b). We found that *Atoh1*[Cre/+]*;Vglut2*[fl/fl] mice were less proficient at this reflex and had a longer turn latency than control littermates at P7 and P11 (Fig. 3b). Second, we measured the surface righting reflex by placing mice on their backs and measuring the time it took for them to turn onto their four paws (Fig. 3c). We found that *Atoh1*[Cre/+]*;Vglut2*[fl/fl] mice were severely impaired in this reflex and required a longer time to turn than littermates at all timepoints measured (Fig. 3c, P7-P11). Finally, we measured social vocalizations in a social isolation paradigm. When separated from the nest, pups vocalize to attract attention from their

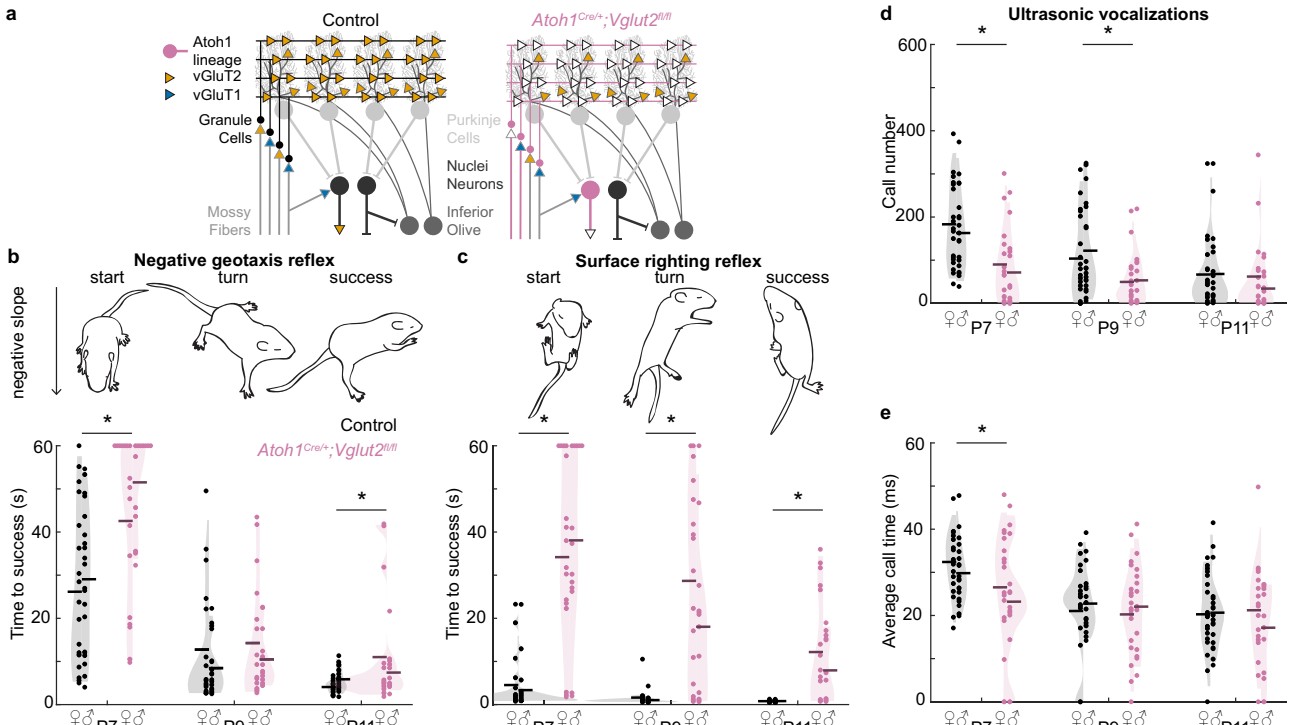

**Fig. 3 | Motor behavior and social behavior are abnormal in early postnatal *Atoh1*^Cre/+^;*Vglut2*^fl/fl^ mice. a** Schematic of circuit modifications in *Atoh1*^Cre/+^;*Vglut2*^fl/fl^ conditional knockout mice. VGluT1 (blue); VGluT2 (orange); Control mice (black); *Atoh1*^Cre/+^;*Vglut2*^fl/fl^ mice (reddish purple). **b** The time to turn upward on a negative slope was measured. *Atoh1*^Cre/+^;*Vglut2*^fl/fl^ conditional knockout mice required a longer time to turn compared to control littermates at P7 ($p < 0.001$; $d = 0.97$) and P11 ($p = 0.015$; $d = 0.59$) but not at P9 ($p = 0.431$; $d = 0.19$). **c** The time to right themselves onto their four paws was measured. *Atoh1*^Cre/+^;*Vglut2*^fl/fl^ mice required a longer time to turn compared to control littermates at P7 ($p < 0.001$; $d = 1.49$), P9 ($p < 0.001$; $d = 1.26$), and P11 ($p < 0.001$; $d = 1.09$). **d** The number of vocalizations after separation from the nest was measured. *Atoh1*^Cre/+^;*Vglut2*^fl/fl^ conditional knockout pups made fewer vocalizations than their control littermates at P7 ($p < 0.001$; $d = 0.90$) and P9 ($p = 0.004$; $d = 0.69$) but not at P11 ($p = 0.373$; $d = 0.20$). **e** The duration of vocalizations after separation from the nest was measured.

*Atoh1*^Cre/+^;*Vglut2*^fl/fl^ pups made shorter vocalizations compared to control littermates at P7 ($p = 0.024$; $d = 0.55$), but not at P9 ($p = 0.700$; $d = 0.10$) or P11 ($p = 0.677$; $d = 0.10$). Control: $N = 38$ (18f/20m); *Atoh1*^Cre/+^;*Vglut2*^fl/fl^: $N = 30$ (17f/13m). Dots represent the means for each mouse, horizontal lines represent the group means, and shaded areas represent the distributions of the data. Data points from control mice in black, and data points from *Atoh1*^Cre/+^;*Vglut2*^fl/fl^ mice in reddish purple. A linear mixed model analysis with genotype as a fixed variable and mouse number as a random variable was used to test for statistical significance in (**b–e**). A post hoc analysis was performed to test for the statistical differences at each time point. Source data and detailed statistical results are available and provided as a Source Data file. Panels (**b**, **d**) were adapted from Van der Heijden et al., 2022, "Quantification of Behavioral Deficits in Developing Mice With Dystonic Behaviors," Dystonia[47] under CC BY 4.0.

mother. We found that *Atoh1*^Cre/+^;*Vglut2*^fl/fl^ mice made fewer calls at P7 and P9 (Fig. 3d), and their calls were shorter at P7 (Fig. 3e). Altogether, our results confirmed that eliminating glutamatergic neurotransmission from *Vglut2*-expressing *Atoh1* lineage neurons impaired the acquisition of reflexive motor functions and social vocalizations in P7–P11 mice.

## The *Atoh1-Vglut2* intersectional lineage defines a population of midbrain-projecting nuclei neurons and other pre-cerebellar nuclei

Previous studies have shown that social behavior may be modulated by the cerebellum through direct projections from glutamatergic cerebellar nuclei to the ventral tegmental area (VTA)[48] and disynaptic connectivity via the ventral posteromedial thalamus (VPM) to the prelimbic area in the medial prefrontal cortex[36]. Other studies have also suggested that these long-range projections from glutamatergic cerebellar nuclei neurons may mediate non-motor behaviors[27,49]. We therefore tested whether the *Vglut2*-expressing, *Atoh1* lineage neurons make long-range projections to areas previously implicated in non-motor behavior. To investigate the complete population of neurons that were affected by our manipulation, we used an intersectional reporter allele (*Rosa*^fsf-lsl-tdTomato^) that expresses tdTomato only after FlpO- (*Atoh1*^FlpO/+^) and Cre- (*Vglut2*^IRES-Cre^) mediated excision of two stop cassettes. In these

mice, only the neurons with a history of *Atoh1* and *Vglut2* expression express tdTomato (Fig. 4). We found tdTomato⁺ projections in all regions known to receive strong inputs from the cerebellar nuclei and which were previously implicated in cerebellar-dependent non-motor function, including the VPM (Fig. 4a, b) and the VTA (Fig. 4e). We also observed projections to other brain regions that are important for motor behavior, in areas that are known to receive strong inputs from cerebellar nuclei neurons, including the zona incerta (ZI) (Fig. 4c) and red nucleus (Fig. 4d, e). Furthermore, we confirmed the presence of tdTomato⁺ neurons in the granule cell layer and cerebellar nuclei (Fig. 4g–j). We also found tdTomato⁺ axons in the superior and middle cerebellar peduncles, which are the white matter tracts through which cerebellar nuclei neurons send projections to these brain regions (Fig. 4f). We provide high-power images in the supplemental material for all brain regions with identified tdTomato⁺ axons or cell bodies (Supplementary Fig. 6). In conclusion, we demonstrated that our genetic manipulation affected neurons with glutamatergic projections that originate from *Atoh1* lineage neurons and terminate in brain regions that are involved in motor function and social behaviors. Thus, we showed that in the *Atoh1* lineage, *Vglut2*-expressing neurons project directly to brain regions that have been previously implicated as key cerebellar targets for mediating diverse behaviors.

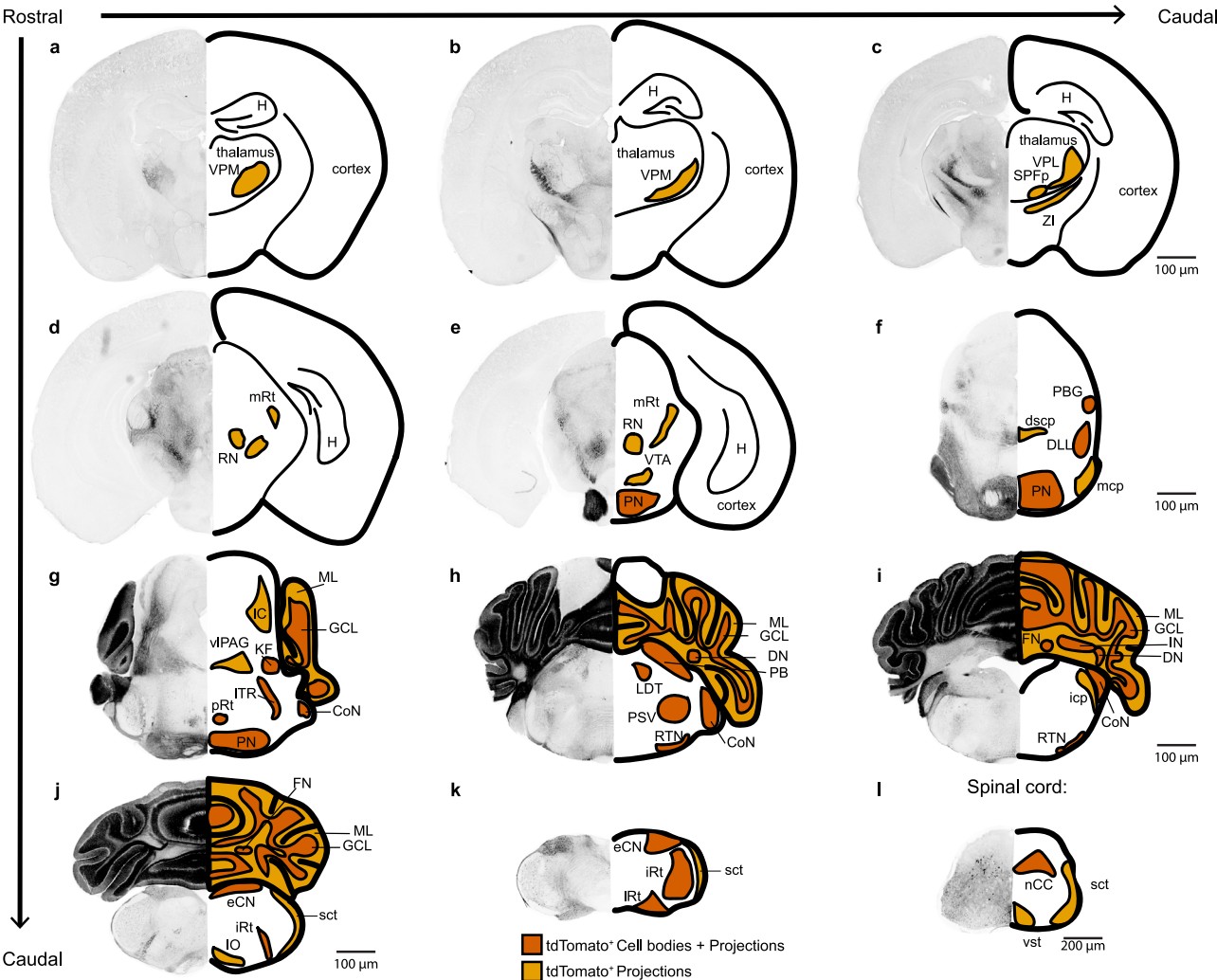

**Fig. 4 | Intersectional lineage labeling of *Atoh1*, *Vglut2* neurons.** tdTomato⁺ neuron labeling in *Atoh1^{FlpO/+}*;*Vglut2^{IRES-Cre/+}*;*Rosa26^{fsf-lsl-tdTomato}* mice. Cell bodies are shown in dark orange, projections are shown in light orange. Abbreviations: CoN cochlear nucleus, dscp decussation of dorsal superior cerebellar peduncle, DLL dorsal lateral lemniscus, DN dentate nucleus, eCN external cuneate nucleus, FN fastigial nucleus, GCL granule cell layer, H Hippocampus, IC inferior colliculus, icp inferior cerebellar peduncle, IN interposed nucleus, IO inferior olive, iRt intermediate reticular nucleus, ITR intertrigeminal region, KF Kölliker Fuse, LDT lateral dorsal tegmental nucleus, lRt lateral reticular nucleus, mcp medial cerebellar peduncle, mRt midbrain reticular nucleus, nCC non-Clark's column, PB parabrachial nucleus, PBG parabigeminal nucleus, PN pontine nuclei, pRt pontine reticular nucleus, PSV principal sensory nucleus of the trigeminal, RN red nucleus, RTN retrotrapezoid nucleus, sct spinocerebellar tract, SPFp subparafascicular nucleus, parvicellular part, vlPAG ventral lateral periaqueductal gray, VPM ventral posteromedial nucleus of the thalamus, VPL ventral posterolateral nucleus of the thalamus, vst ventral spinothalamic tract, VTA ventral tegmental area, ZI zona incerta. Brain images in (a–k) are shown at the same magnification. The spinal cord image in (l) is shown at a different magnification. The light orange brain regions mean that only tdTomato⁺ projections were observed, and the vermillion brain regions mean the cell bodies (and the projections) were observed. Images are representative of *N* = 3 mice.

## Granule cells express *Vglut1*, but not *Vglut2*, in adult mice

Next, we wanted to investigate whether some of the behavioral deficits observed in the early postnatal *Atoh1^{Cre/+}*;*Vglut2^{fl/fl}* mice (Fig. 3) would be restored in adult mice after the granule cells, but not nuclei neurons, switch their transporter subtype from *Vglut2* to *Vglut1* (Fig. 5a–c). We used in situ histochemistry to confirm that *Atoh1*-lineage granule cells, but not nuclei neurons, express *Vglut1* in adult mice (Fig. 5d). We further show that in control and *Atoh1^{Cre/+}*;*Vglut2^{fl/fl}* conditional knockout mice, the vesicular transporter subtype switching only occurred in granule cells, but not nuclei neurons (Fig. 5e). Finally, we confirm that the expression of VGluT1 in the molecular layer of crus I and lobule V in P7 (Fig. 5f) and adult (Fig. 5g) is not different between control and *Atoh1^{Cre/+}*;*Vglut2^{fl/fl}* mice. Together these results confirm that a molecular switch from *Vglut2* to *Vglut1* expression in granule cells occurs after early postnatal development, that this switch is independent of our genetic

manipulation, and that our genetic manipulation does not drive a similar molecular switch in glutamatergic cerebellar nuclei neurons (Fig. 5).

## Purkinje cells in adult *Atoh1^{Cre/+}*;*Vglut2^{fl/fl}* mice have normal in vivo firing activity

There is experimental evidence showing that restoring cerebellar function during development may be sufficient to rescue social deficits in mouse models for autism spectrum disorders[50]. We, therefore, set out to investigate whether the natural developmental transition from VGluT2- to VGluT1-mediated neurotransmission in granule cells (Fig. 5) is associated with the normalization of Purkinje cell firing rates, representing the restoration of normal cerebellar cortical function in adult *Atoh1^{Cre/+}*;*Vglut2^{fl/fl}* mice. When we performed single-unit recordings of Purkinje cells in the adult mutant mice (Fig. 6), we did not observe differences in the simple spike or

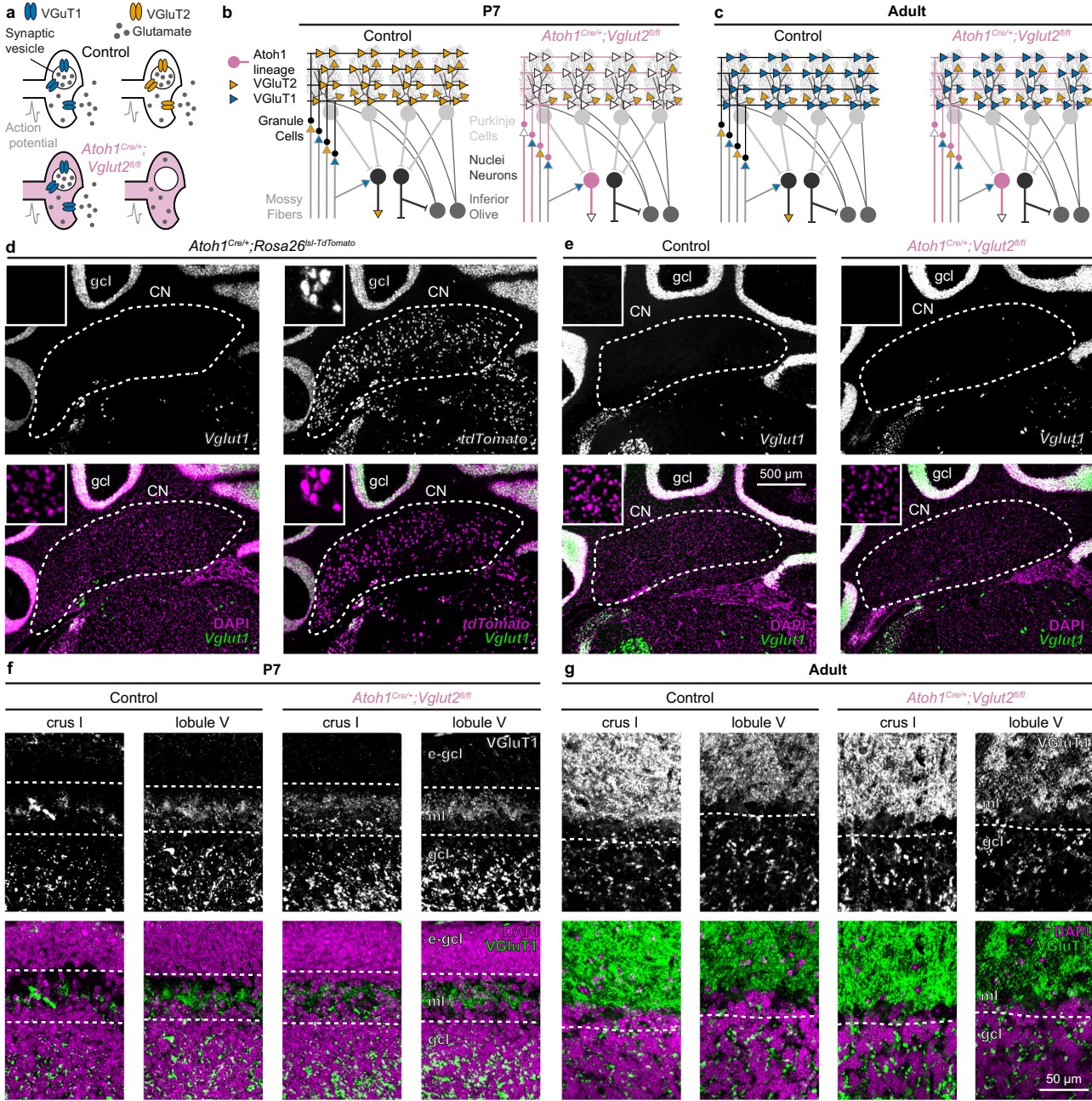

**Fig. 5 | *Vglut1* expression in control and *Atoh1^(Cre/+)*;*Vglut2^(fl/fl)* mice. a** Schematic showing how conditional deletion of VgluT2 does not affect fast neurotransmission in VgluT1-expressing neurons. **b** Schematic of cerebellar connectivity and vesicular transporter expression for the glutamate subtypes (VgluT1 and VgluT2) in the cerebellar circuit in P7 control mice and *Atoh1^(Cre/+)*;*Vglut2^(fl/fl)* conditional knockout mice. **c** Same as **b**. but in adult mice. For **a–c** VgluT1 (blue); VgluT2 (orange); Control mice (black); *Atoh1^(Cre/+)*;*Vglut2^(fl/fl)* mice (reddish purple). **d** Expression of *Vglut1* (green) with DAPI (purple, left) or *tdTomato* (purple, right) in the cerebellar nuclei (CN) and granule cell layer (gcl) of adult *Atoh1^(Cre/+)*;*Rosa26^(lsl-tdTomato)* mice. **e** Expression of *Vglut1* (green) and DAPI (purple) in the CN and gcl of adult control mice and *Atoh1^(Cre/+)*;*Vglut2^(fl/fl)* mice. **d**, **e** Insets are 125 by 125 μm high magnification images. **f** VgluT1^+ synapses (green) and DAPI (purple) in the molecular layer (ml) and gcl of P7 control and *Atoh1^(Cre/+)*;*Vglut2^(fl/fl)* mice, e-gcl = external granule cell layer containing actively proliferating granule cell precursor cells. **g** VgluT1^+ synapses (green) and DAPI (purple) in the ml and gcl of adult control mice and *Atoh1^(Cre/+)*;*Vglut2^(fl/fl)* mice. **d**, **e** are shown at the same scale. **f**, **g** are shown at the same scale. Images are representative of *N* = 3 mice.

complex spike firing rate, pattern, or regularity. These findings showed that even though *Vglut2* elimination from granule cells during the developmental period initially caused an abnormal simple spike firing rate (Fig. 2d), the Purkinje cell firing rate later normalized after granule cells switched their transporter type to *Vglut1*, which persists into adulthood (Fig. 6). The occurrence of Purkinje cell firing rate normalization supports the prediction that in adults, the glutamatergic cerebellar nuclei neurons, but not neurons that are located upstream in the cerebellar cortex, are functionally affected by VGluT2 loss.

## Adult *Atoh1^(Cre/+)*;*Vglut2^(fl/fl)* mice have altered motor function but no social behavior deficits

Next, we set out to determine whether the normalization of cerebellar cortical function was paired with the restoration of cerebellar-dependent behaviors (Fig. 7). We measured motor function in adult mice using assays that reveal behavioral impairments in mouse models with cerebellar deficits. First, we measured tremors. We previously found that eliminating GABAergic neurotransmission from Purkinje cells reduces tremor[51], whereas eliminating glutamatergic neurotransmission from climbing fibers or impairing the

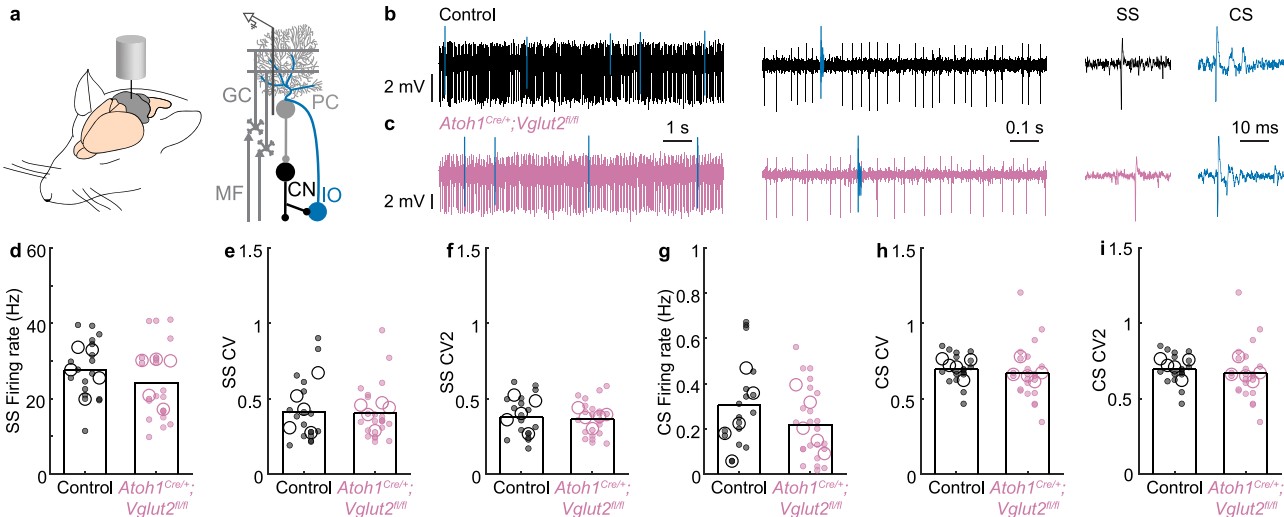

**Fig. 6 | Purkinje cell firing activity in adult mice. a** Schematic of the in vivo single-unit extracellular Purkinje cell recording setup in anesthetized mice.
**b** Representative trace of a Purkinje cell recording in a control mouse (black and blue lines represent complex spikes). **c** Representative trace of a Purkinje cell recording in an $Atoh1^{Cre/+};Vglut2^{fl/fl}$ mouse (reddish purple, blue lines represent complex spikes). In **b, c,** Inferior olive (IO) evoked a complex spike (CS) in blue. The *y*-axis is constant across the panel. The *x*-axes (timescales) are the same for (**b, c**). **d** No differences were found in the SS Firing rate ($p = 0.490$; $d = 0.40$), **e** SS CV (spike pattern, $p = 0.766$; $d = 0.07$), **f** SS CV2 (spike regularity, $p = 0.659$; $d = 0.11$), **g** CS Firing rate ($p = 0.489$, $d = 0.49$), **h** CS CV (spike pattern, $p = 0.598$, $d = 0.19$), or

**i** CS CV2 (spike regularity, $p = 0.246$, $d = 0.48$). For **d–i**, large open circles represent the mouse average, and small, closed circles represent the cell average. Data points from control mice in black, and data points from $Atoh1^{Cre/+};Vglut2^{fl/fl}$ mice in reddish purple. A linear mixed model analysis with genotype as a fixed variable and mouse number as a random variable was used to test for statistical significance in (**d–i**). Control: $N = 5$ mice, $n = 18$ cells; $Atoh1^{Cre/+};Vglut2^{fl/fl}$: $N = 5$, $n = 23$. Source data and detailed statistical results are available and provided as a Source Data file. Panel (**a**) was adapted from White & Sillitoe, 2017, "Genetic silencing of olivocerebellar synapses causes dystonia-like behavior in mice," Nature Communications[21] under CC BY 4.0.

neurogenesis of *Atoh1* neurons increases tremor power in mice[29,52]. Changes in tremor intensity are thus a reliable, functional readout for assessing cerebellar neuron manipulations in freely behaving mice. We found that the $Atoh1^{Cre/+};Vglut2^{fl/fl}$ conditional knockout mice have a modest reduction in the average, but not peak, tremor power compared to their control littermates (Fig. 7a–c). This reduction in tremor power is similar to what has been observed previously in mice lacking GABAergic neurotransmission from Purkinje cells[51].

Second, we tested the ambulatory activity of the conditional knockouts in the open field assay, as alterations in cerebellar circuit function often result in changes in ambulatory activity in the open field[29,52,53]. Indeed, we observed that compared to control littermates, the $Atoh1^{Cre/+};Vglut2^{fl/fl}$ mice move less often (Fig. 7d), although the total distance traveled is not different (Fig. 7e), and they have fewer horizontal and vertical activity events (Fig. 7f, g). These measurements show that the adult $Atoh1^{Cre/+};Vglut2^{fl/fl}$ conditional knockout mice initiate fewer spontaneous motor events in an open field.

Third, we tested whether motor performance and learning are impaired in the adult $Atoh1^{Cre/+};Vglut2^{fl/fl}$ mice. We tested the latency to fall on an accelerating rotarod as this behavior is often impaired in mice with cerebellar deficits[21,42,53,54]. We indeed found that motor performance and motor learning are impaired in $Atoh1^{Cre/+};Vglut2^{fl/fl}$ mice when compared to control littermates (Fig. 7h). Altogether, cerebellar-dependent motor function is impaired in adult $Atoh1^{Cre/+};Vglut2^{fl/fl}$ mice when fast neurotransmission from glutamatergic nuclei, but not granule cells, is genetically eliminated.

Fourth, we tested whether, despite these collective impairments in motor function, social behaviors were restored after the normalization of cerebellar cortical function by the transporter switch. We tested sociability in the three-chamber assay because previous studies have shown that sociability in this assay is impaired in many mouse models with cerebellar deficits[36,48,55]. We did not observe differences in the social approach assay in $Atoh1^{Cre/+};Vglut2^{fl/fl}$ conditional knockout mice compared to their control littermates (Fig. 7i). Together, our findings demonstrate that while chronic alteration of fast

neurotransmission from *Atoh1* lineage glutamatergic cerebellar nuclei neurons continues to impair the proper execution of motor functions into adulthood, it still permits the restoration of social behaviors associated with the normalization of cerebellar cortical function.

### The $Ntsr1^{Cre}$ and $Vglut2$ co-expressing neurons define a subset of glutamatergic cerebellar nuclei neurons

Based on the behavioral and electrophysiological findings in the $Atoh1^{Cre/+};Vglut2^{fl/fl}$ mice, we next wondered whether the early social deficits and delayed acquisition of normal social behaviors in adult $Atoh1^{Cre/+};Vglut2^{fl/fl}$ mice were solely mediated by the progressive normalization of cerebellar cortical function. If so, then selectively silencing the neurotransmission from glutamatergic cerebellar nuclei neurons while leaving cerebellar cortical function intact would be predicted to influence the acquisition of early motor behaviors but not social vocalizations. To address this question, we examined the developmental contribution of $Ntsr1^{Cre}$ expressing cerebellar nuclei neurons towards the acquisition of behaviors during development. Previous studies have shown that these glutamatergic neurons send axonal projections to the thalamus and VTA, among other midbrain and brainstem regions[28,34,56].

In agreement with previous studies, we found $Ntsr1^{Cre}$-driven *YFP* expression in the cerebellar nuclei neurons (Fig. 8a inset iii). We also found *YFP* expression in layer 6 neurons of the cerebral cortex (Fig. 8a inset i) and sparse labeling in the striatum (Fig. 8a inset iv), midbrain, (Fig. 8a inset v), and medulla (Fig. 8a inset vi). While there was some *Vglut2* expression in overlapping areas of the midbrain and medulla, we did not observe any co-expression of *YFP* and *Vglut2* in the same neurons in any brain region other than the cerebellar nuclei (Fig. 8a inset iii).

We found $Ntsr1^{Cre}$-driven *YFP* expression in some, but not all, $Vglut2^+$ cerebellar nuclei neurons (Fig. 8a–c). Additionally, $Ntsr1^{Cre}$-driven *YFP* expression was not homogeneous across cerebellar nuclei (Fig. 8b, c). We counted $YFP^+$ and $Vglut2^+$ neurons in the dentate, interposed, and fastigial nuclei within the cerebellum (Fig. 8b, c). We

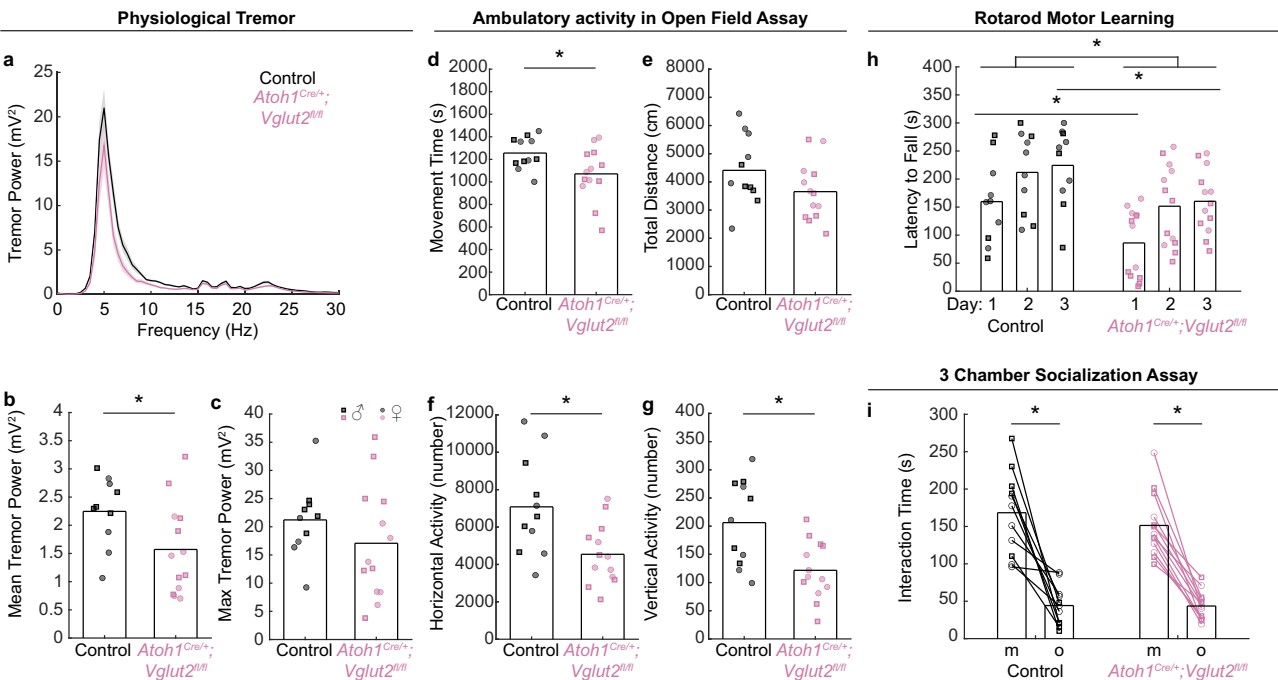

**Fig. 7 | Motor behavior and social behavior in adult *Atoh1*[Cre/+]*;Vglut2*[fl/fl] conditional knockout mice. a–c** A tremor in freely moving mice. Control: $N = 10$ (5 female/5 male); *Atoh1*[Cre/+]*;Vglut2*[fl/fl]: $N = 13$ (6 f/7 m). **a** Power spectrum of tremor in control and *Atoh1*[Cre/+]*;Vglut2*[fl/fl] conditional knockout mice. **b** Mean tremor power between 0 and 30 Hz was lower in *Atoh1*[Cre/+]*;Vglut2*[fl/fl] mice compared to control mice ($p = 0.039$; $d = 0.90$). **c** Maximum tremor power was not different between *Atoh1*[Cre/+]*;Vglut2*[fl/fl] conditional knockouts and control mice ($p = 0.276$; $d = 0.63$). **d–g** Ambulatory activity in the open field assay. Control: $N = 11$ (6 f/5 m); *Atoh1*[Cre/+]*;Vglut2*[fl/fl]: $N = 13$ (6 f/7 m). *Atoh1*[Cre/+]*;Vglut2*[fl/fl] mice had lower (in **d**) total movement time ($p = 0.03$; $d = 0.86$), **f** horizontal activity count ($p = 0.008$; $d = 1.03$), and **g** vertical activity count ($p = 0.004$; $d = 1.11$) compared to control mice, but no difference in (**e**) total distance traveled ($p = 0.116$; $d = 0.65$). **h** Rotarod performance. Control: $N = 10$ (6 f/4 m); *Atoh1*[Cre/+]*;Vglut2*[fl/fl]: $N = 13$ (6 f/7 m). *Atoh1*[Cre/+]*;Vglut2*[fl/fl] mice fell off the rotarod faster than control mice ($p = 0.017$), but the group difference was

only significant on the first day ($p = 0.016$; $d = 0.83$) and third training day ($p = 0.025$; $d = 0.73$), but not the second training day ($p = 0.062$; $d = 0.58$). **i** Three-chamber socialization assay. Control: $N = 11$ (6 f/5 m); *Atoh1*[Cre/+]*;Vglut2*[fl/fl]: $N = 13$ (6 f/7 m). There was no difference between *Atoh1*[Cre/+]*;Vglut2*[fl/fl] mice and control mice in the total time spent with a mouse ($p = 0.404$; $d = 0.08$) or an object ($p = 0.948$; $d = 0.13$), and both groups spent more time with the mouse than the object (control: $p < 0.001$; *Atoh1*[Cre/+]*;Vglut2*[fl/fl]: $p < 0.001$ in a two-sided paired *t*-test). A two-sided unpaired *t*-test was used to test for statistical significance unless otherwise noted. Black circles and squares represent data points from control mice; reddish purple circles and squares represent data points from *Atoh1*[Cre/+]*;Vglut2*[fl/fl] mice. Squares represent data points from male mice, circles represent data points from female mice. Source data and detailed statistical results are available and provided as a Source Data file.

found that more than half of *Vglut2*[+] neurons in the dentate nucleus were also *YFP*[+] (56.6 ± 0.01 %), nearly three-quarters of *Vglut2*[+] neurons in the interposed nucleus were also *YFP*[+] (73.7 ± 1.3%), and around a sixth of *Vglut2*[+] neurons in the fastigial nucleus were also *YFP*[+] (16.5 ± 1.6%). Overall, around half of all *Vglut2*[+] cerebellar nuclei neurons were also *YFP*[+] (52.5 ± 1.1%) (Fig. 8).

Next, we set out to confirm that these glutamatergic, *Ntsr1*[Cre]-expressing nuclei neurons send long range projections to areas associated with motor and non-motor behaviors. To this end, we crossed the *Ntsr1*[Cre] mice to *Atoh1*[FlpO/+]*;Rosa*[fsf-lsl-tdTomato] mice to obtain mice that expressed tdTomato solely in the *Ntsr1*[Cre] expressing cerebellar nuclei neurons (Supplementary Fig. 7). We confirmed that the tdTomato is expressed in the cerebellar nuclei neurons in a pattern that is similar to the *YFP* expression in the in situ experiments (Supplementary Fig. 7a). Specifically, we found the highest density of tdTomato[+] neurons in the interposed nuclei, some tdTomato[+] neurons in the dentate nuclei, and a few tdTomato[+] neurons in the fastigial nuclei (Supplementary Fig. 7b). We also see some tdTomato[+] projections traveling to the molecular layer within the cerebellar cortex (Supplementary Fig. 7b), which may explain the presence of fiber tracks in the cerebellar cortex observed in a previous description of this mouse line[57]. Additionally, we observed tdTomato[+] projections in the thalamus, red nucleus, midbrain, and brainstem (Supplementary Fig. 7c–k). This includes some sparse labeling of projections in regions of TH[+] VTA neurons and in the inferior olive, which is in agreement with previous studies[28,48]. Together, these studies show that *Ntsr1*[Cre]-expressing cerebellar nuclei

neurons are integrated into circuits that were previously associated with motor and non-motor cerebellar-dependent behaviors.

## Early postnatal *Ntsr1*[Cre/+]*;Vglut2*[fl/fl] mice exhibit motor deficits with normal social behavior

We next sought to test whether *Vglut2*-dependent fast neurotransmission from these *Ntsr1*[Cre] cerebellar nuclei neurons is required for the acquisition of motor function and social vocalizations during the postnatal period in mice. To do so, we generated *Ntsr1*[Cre]*;Vglut2*[fl/fl] conditional knockout mice to eliminate glutamatergic neurotransmission from a select population of the *Ntsr1*[Cre] cerebellar nuclei neurons (Fig. 9a).

We tested whether the *Ntsr1*[Cre]*;Vglut2*[fl/fl] mice have abnormal motor behavior in the early postnatal period by testing the integrity of their postnatal motor reflexes. We found that similar to the *Atoh1*[Cre]*;Vglut2*[fl/fl] mice, *Ntsr1*[Cre]*;Vglut2*[fl/fl] mice have a longer latency to turn upward in the negative geotaxis reflex compared to control littermates on P7, P9, and P11 (Fig. 9b). Additionally, the *Ntsr1*[Cre]*;Vglut2*[fl/fl] mice also have a longer latency to turn onto their four paws in the righting reflex when compared to control littermates on P7, P9, and P11 (Fig. 9c). These findings altogether confirm that fast glutamatergic neurotransmission from the cerebellar nuclei neurons is essential for the proper acquisition of motor behavior during postnatal development in mice.

We also examined the composition of pup vocalizations in a social isolation paradigm. Surprisingly, we did not observe fewer or shorter

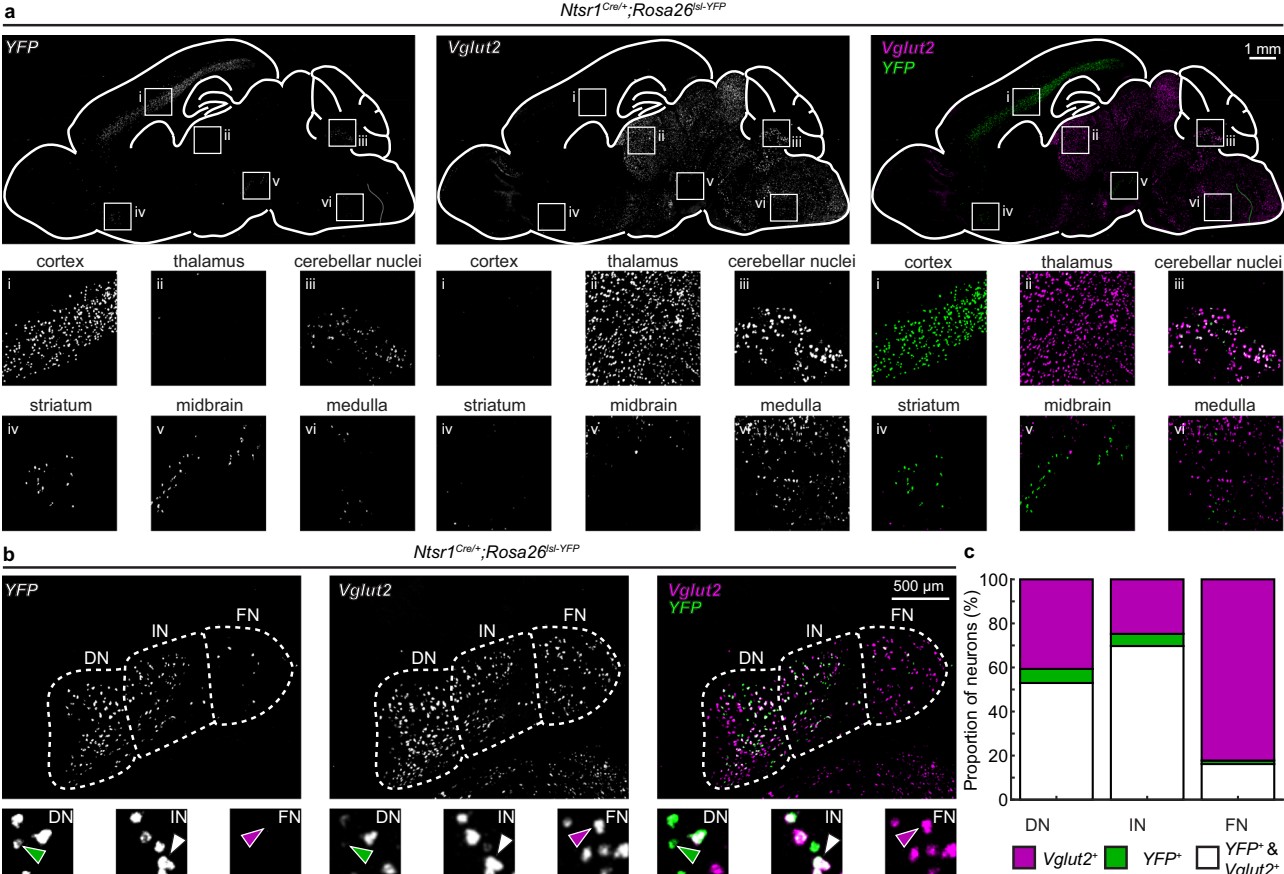

**Fig. 8 | *Ntsr1^Cre* marks a subset of glutamatergic cerebellar nuclei neurons.**
**a** Expression of *Ntsr1^Cre* mapped through Cre-dependent *YFP* expression and *Vglut2* in sagittal brain slices. Insets show high magnification of *YFP* and *Vglut2* expression in (i) the cerebral cortex, (ii) the thalamus, (iii) cerebellar nuclei, (iv) the striatum, (v) the midbrain, and (vi) the medulla. We only observe co-expression of *YFP* (green, right panels) and *Vglut2* (purple, right panels) in the same neurons in the cerebellar nuclei. **b** Expression of *Ntsr1^Cre* mapped through Cre-dependent *YFP* expression (green) in glutamatergic (*Vglut2* expressing, purple) cerebellar nuclei neurons

(coronal sections). The green arrowhead indicates a *YFP*, but not *Vglut2*, expressing neuron; the white arrowhead indicates a *YFP* and *Vglut2* expressing neuron; and the purple arrowhead indicates a *Vglut2*, but not *YFP*, expressing neuron.
**c** Quantification of *Vglut2^+* (purple), *YFP^+* (green), and *Vglut2^+* & *YFP^+* double-positive neurons (white) in the DN, IN, and FN. $N = 3$ mice, $n = 6$ nuclei (3 brain sections) per mouse. Source data are provided as a Source Data file. DN dentate nucleus, IN interposed nucleus, FN fastigial nucleus. Images are representative of $N = 3$ mice.

vocalizations in the *Ntsr1^Cre*;*Vglut2^fl/fl* conditional knockout mice compared to control littermates (Fig. 9d, e), which is in contrast to what we observed in age-matched *Atoh1^Cre/+*;*Vglut2^fl/fl* conditional knockout mice (Fig. 3d, e). We did observe a greater number of vocalizations in the *Ntsr1^Cre*;*Vglut2^fl/fl* mice at P11 (Fig. 9d), which was not observed in control littermates or age-matched *Atoh1^Cre/+*;*Vglut2^fl/fl* conditional knockout mice (Fig. 3d). These observations suggest that early postnatal glutamatergic neurotransmission from the cerebellar nuclei neurons is not essential for the acquisition of social vocalizations as assessed in 2-week-old mice.

### Adult *Ntsr1^Cre/+*;*Vglut2^fl/fl* mice do not exhibit motor dysfunction or social behavior deficits
Finally, we set out to investigate whether the motor phenotypes observed in the early postnatal *Ntsr1^Cre/+*;*Vglut2^fl/fl* mice would remain impaired or be restored by adulthood. These experiments would unveil whether this subset of glutamatergic cerebellar nuclei neurons is essential for the acquisition of motor behaviors throughout life, or if developmental compensation may occur over time. It also provides an opportunity to partially disentangle how chronic impairment of only the glutamatergic cerebellar nuclei, but not the cerebellar cortex (as in *Atoh1^Cre/+*;*Vglut2^fl/fl* mice), may affect adult behaviors. We found that compared to control mice, adult *Ntsr1^Cre/+*;*Vglut2^fl/fl* mice had no deficits in ambulatory activity (Fig. 10a–d), rotarod motor learning

(Fig. 10e), or social preference in the 3-chamber assay (Fig. 10f). Thus, despite severe early postnatal deficits in motor reflexes (Fig. 9), adult *Ntsr1^Cre/+*;*Vglut2^fl/fl* mice show no changes in motor function. These results underscore the remarkable ability of the brain to compensate for some developmental perturbations in cerebellar function.

## Discussion
In this study, we combined intersectional mouse genetics and age-appropriate behavioral paradigms to investigate how perturbing the function of neurons in the cerebellar cortex and cerebellar nuclei causes acute and prolonged motor dysfunction and social deficits. We found that developmental elimination of *Vglut2*-dependent glutamatergic neurotransmission from *Atoh1* lineage neurons severely impacts motor behavior and social vocalizations in the early postnatal period (Fig. 10g). By the time these *Atoh1^Cre/+*;*Vglut2^fl/fl* mutant mice reach adulthood, the natural *Vglut2* to *Vglut1* transporter switch within granule cells and the concomitant normalization of cerebellar cortical function coincide with normal social behaviors and only mild motor deficits. When we eliminated neurotransmission from *Ntsr1* expressing glutamatergic nuclei neurons, we did not observe social deficits but did detect severe motor deficits in early postnatal mice that fully resolved with age. Together, we found that glutamatergic neurotransmission from cerebellar cortical versus cerebellar nuclei neurons differentially control the acquisition of motor and social behaviors and

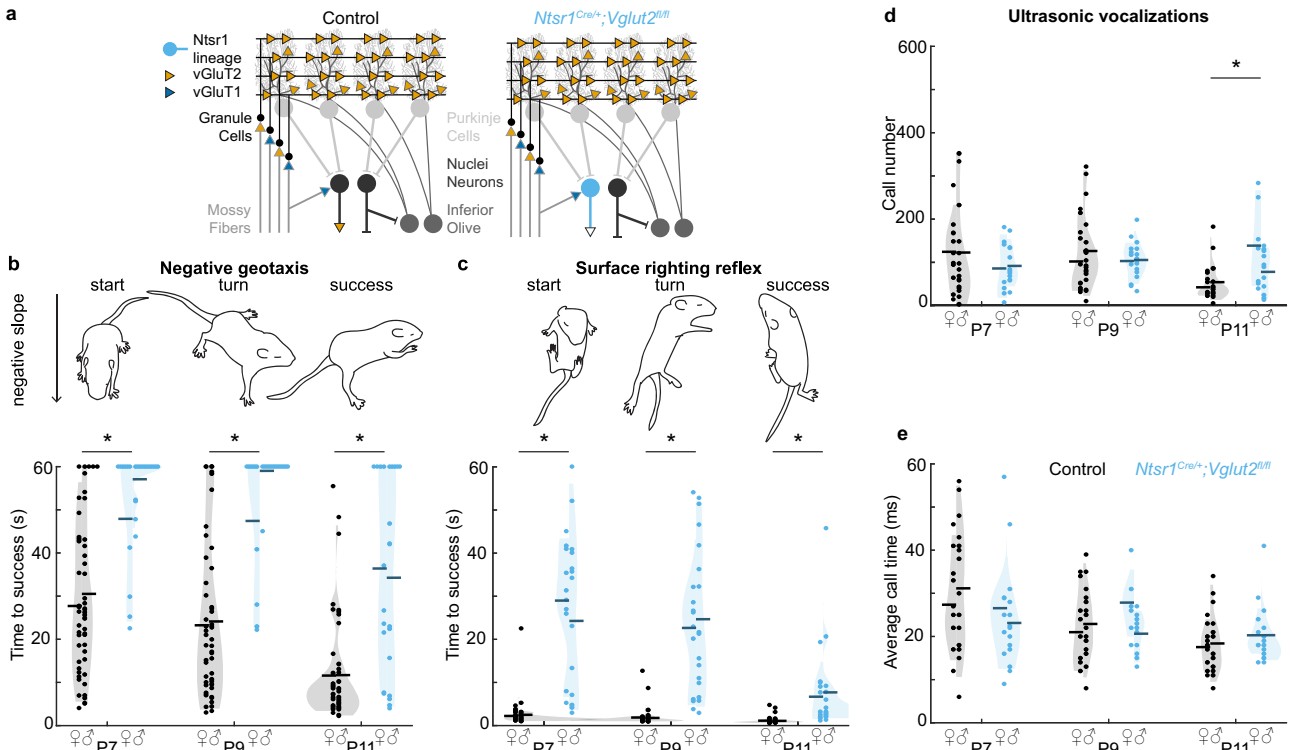

**Fig. 9 | Motor behavior and social behavior in early postnatal *Ntsr1^Cre^;Vglut2^fl/fl*** **mice. a** Schematic of circuit modifications in *Ntsr1^Cre/+^;Vglut2^fl/fl^* conditional knockout mice. VGluT1 (blue); VGluT2 (orange); Control mice (black); *Ntsr1^Cre/+^;Vglut2^fl/fl^* mice (sky blue). **b** Time to turn upward on a negative slope was measured. *Ntsr1^Cre/+^;Vglut2^fl/fl^* mice required a longer time to turn compared to control littermates at P7 ($p < 0.001$; d = 1.34), P9 ($p < 0.001$; $d = 1.43$), and P11 ($p < 0.001$; $d = 1.91$). **c** Time to right onto four paws was measured. *Ntsr1^Cre/+^;Vglut2^fl/fl^* mice required a longer time to turn compared to control littermates at P7 ($p < 0.001$; $d = 2.13$), P9 ($p < 0.001$; $d = 3.63$), and P11 ($p < 0.001$; $d = 3.53$). **d** Number of vocalizations after separation from the nest was measured. *Ntsr1^Cre/+^;Vglut2^fl/fl^* conditional knockout pups made a similar number of vocalizations as control littermates at P7 ($p < 0.205$; $d = 0.38$), and P9 ($p = 0.553$; $d = 0.18$), but made more calls at P11 ($p = 0.003$; $d = 0.87$). **e** Duration of vocalizations after separation from the nest was measured. *Ntsr1^Cre/+^;Vglut2^fl/fl^* conditional knockouts made calls of the same duration as

compared to control littermates at P7 ($p = 0.161$; $d = 0.42$), P9 ($p = 0.576$; $d = 0.17$), and P11 ($p = 0.240$; $d = 0.35$). For **b** and **c**, Control: $N = 54$ (22 f/32 m); *Ntsr1^Cre/+^; Vglut2^fl/fl^*: $N = 25$ (10 f/15 m). For **d**, **e**, Control: $N = 30$ (11 f/19 m); *Ntsr1^Cre/+^;Vglut2^fl/fl^*: $N = 18$ (7 f/11 m). Dots represent means for each mouse, horizontal lines represent group means, and shaded areas represent the distribution of the data. Data points from control mice in black, and data points from *Ntsr1^Cre/+^;Vglut2^fl/fl^* mice in sky blue. A linear mixed model analysis with genotype as a fixed variable and mouse number as a random variable was used to test for statistical significance in (**c**–**e**). A post hoc analysis was performed to test for the statistical differences at each time point. Source data and detailed statistical results are available and provided as a Source Data file. Panels (**b**, **d**) were adapted from Van der Heijden et al., 2022, "Quantification of Behavioral Deficits in Developing Mice With Dystonic Behaviors," Dystonia[47] under CC BY 4.0.

that the brain can compensate for some, but not all, perturbations that occur in the developing cerebellum (Fig. 10g).

## Contribution of glutamatergic cerebellar neurons to social behaviors

Considering the breadth of work that reports on the presence of aberrant social behaviors after different cerebellar manipulations[19,36,48,55], it may be contradictory, at first thought, that social behaviors are not widely affected when glutamatergic cerebellar output nuclei neurons are genetically silenced. We recently found that eliminating neurotransmission at glutamatergic climbing fiber synapses, which terminate extensively on the Purkinje cell dendrites, results in a robust reduction of vocalizations in P7–P11 pups[47]. These data are consistent with previous studies that used developmental or acute changes in Purkinje cells to study abnormal social preference in adult mice[17,19,50,55,58]. However, these studies all induced changes in the main cerebellar cortical output neurons, Purkinje cells, which have extensive connections to both the GABAergic and the glutamatergic neurons in the cerebellar nuclei. Ultimately, from those studies, one cannot differentiate whether each functional subtype of cerebellar nuclei neuron might be equally essential for driving social behaviors.

Several previous studies have directly manipulated cerebellar nuclei neurons and measured the secondary effect on social approach

in adult mice. One study found that exciting cerebellar inputs into the VTA results in increased social approach[48]. Based on slice physiology, Carta and colleagues postulate that these cerebellar inputs to the VTA are likely glutamatergic in nature, but their experimental manipulation did not focus on the neurotransmitter identity of the projecting neurons, which may provide monosynaptic GABAergic or glutamatergic nuclei projections to the VTA[28]. Another study suggested that inhibiting neural activity in the dentate nucleus of the cerebellum can alleviate abnormal social preference in a mouse model for autism[36]. Yet, like the work by Carta and colleagues, Kelly and colleagues used experimental approaches that do not experimentally distinguish the neurotransmitter identity of the affected cell types. Similarly, the inactivation of neurons in the fastigial cerebellar nucleus resulted in altered social interactions in rats[59]. These observations are in support of changes in affect and sociability in children with posterior fossa syndrome and related developmental conditions, who often experience perturbations or injuries to the superior peduncle that contains the bulk of output projections from the GABAergic and glutamatergic cerebellar nuclei neurons[60,61]. Thus, although it has been routinely suspected that the glutamatergic cerebellar nuclei play a role in sociability, the relative contribution of the GABAergic versus the glutamatergic cerebellar nuclei neurons to different cerebellar-dependent behaviors remains unknown.

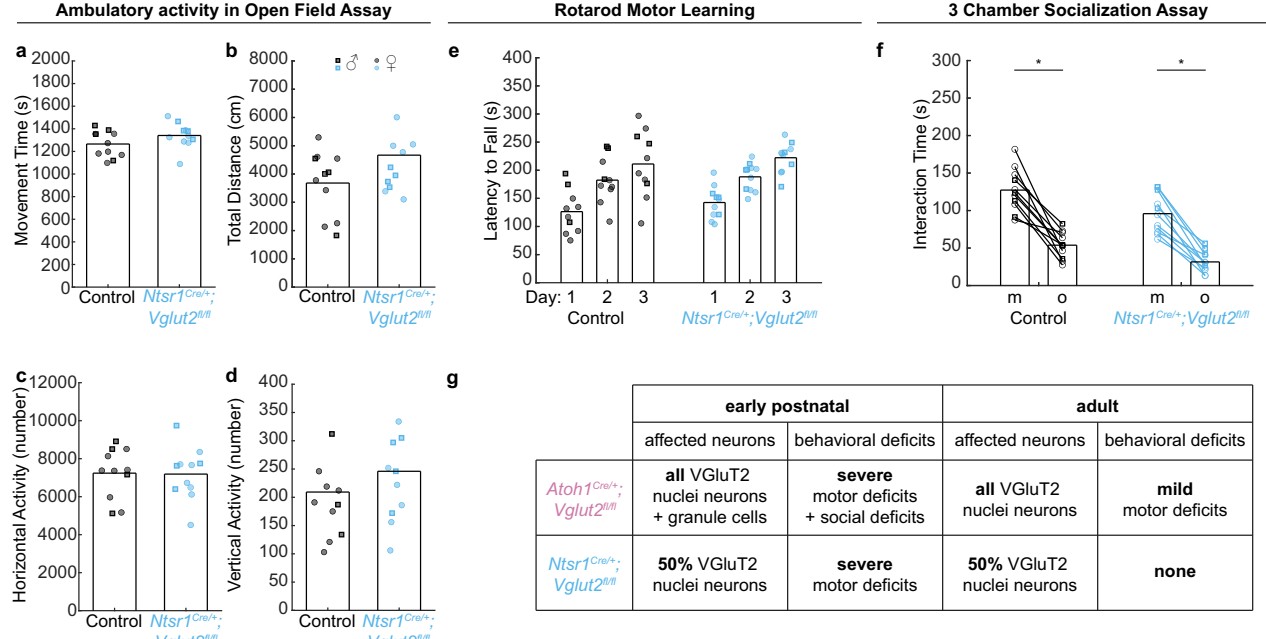

**Fig. 10 | Motor behavior and social behavior in adult *Ntsr1^{Cre/+};Vglut2^{fl/fl}* conditional knockout mice. a–d** Ambulatory activity in the open field assay. Control: $N = 11$ (7 f/4 m); *Ntsr1^{Cre/+};Vglut2^{fl/fl}*: $N = 11$ (7 f/4 m). *Ntsr1^{Cre/+};Vglut2^{fl/fl}* mice were not different from control mice in **a** total movement time ($p = 0.135$; $d = 0.64$), **b** total distance traveled ($p = 0.106$; $d = 0.69$), **c** horizontal activity count ($p = 0.931$; $d = 0.04$), or **d** vertical activity count ($p = 0.410$; $d = 0.41$) compared to control mice. **e** Rotarod performance. Control: $N = 11$ (7 f/3 m); *Ntsr1^{Cre/+};Vglut2^{fl/fl}*: $N = 11$ (7 f/4 m). There was no difference in latency to fall between *Ntsr1^{Cre/+};Vglut2^{fl/fl}* mice and control mice ($p = 0.786$). **f.** 3-chamber socialization assay. Control: $N = 11$ (7 f/4 m); *Ntsr1^{Cre/+};Vglut2^{fl/fl}*: $N = 11$ (7 f/4 m). *Ntsr1^{Cre/+};Vglut2^{fl/fl}* mice spend less time interacting with both the mouse ($p = 0.014$; $d = 1.01$) and the object ($p = 0.004$; $d = 1.15$) than control mice, but both groups spent more time with the mouse than the object (control: $p < 0.001$; *Ntsr1^{Cre/+};Vglut2^{fl/fl}*: $p < 0.001$ in the paired *t*-test). A two-sided unpaired *t*-test was used to test for statistical significance unless otherwise noted. **g** Summary of results. Black circles and squares represent data points from control mice; sky blue circles and squares represent data points from *Ntsr1^{Cre/+};Vglut2^{fl/fl}* mice. Squares represent data points from male mice, circles represent data points from female mice. Source data and detailed statistical results are available and provided as a Source Data file.

Although anatomical studies suggest that the glutamatergic nuclei neurons may contribute to various non-motor functions[27,49,62], we found that eliminating the neurotransmission from most or all glutamatergic nuclei neurons leads to only a temporary social deficit or no observable change in sociability, respectively. Our results suggest, perhaps expectedly, that motor and non-motor functions are mediated through parallel glutamatergic and GABAergic pathways emerging from the cerebellar nuclei. Future studies could selectively modulate signaling from the GABAergic cerebellar nuclei to better investigate whether these neurons have dedicated and/or independent roles that are required for the establishment of cerebellar-dependent social behaviors. Alternatively, the signals needed for social behavior may be encoded by complementary connections with Purkinje cells that have simultaneous and parallel interactions with both the glutamatergic and the GABAergic cerebellar nuclei neurons.

Given these considerations, we propose the following. Social behaviors require normal function and the development of Purkinje cells. When the Purkinje cells do not function normally, they propagate their aberrant signals through downstream glutamatergic and GABAergic cerebellar nuclei neurons. Glutamatergic nuclei neurons may modulate social behaviors when acutely inhibited or excited, but their signaling is not essential for the acquisition of social behaviors because parallel signaling from GABAergic cerebellar nuclei neurons may be sufficient to propagate information regarding social interaction from Purkinje cells to other brain regions[28]. However, our experimental approach does not delineate whether these GABAergic projection neurons are equally sufficient in regulating social behaviors when the glutamatergic neuron function is perturbed after circuit development or maturation. Future studies are needed to further

disentangle the contributions of each nuclear cell type to social behavior throughout life.

## Developmental compensation of motor functions

We found a robust and significant difference in the severity of motor deficits when we compared early postnatal conditional knockout mice to adult *Atoh1^{Cre/+};Vglut2^{fl/fl}* and *Ntsr1^{Cre/+};Vglut2^{fl/fl}* mice. Although we performed our experiments in a blinded fashion, the motor abnormalities in early postnatal *Atoh1^{Cre/+};Vglut2^{fl/fl}* and *Ntsr1^{Cre/+};Vglut2^{fl/fl}* mice were often readily visible as abnormal movements and postures that persist even in the resting state. In contrast, no overt signs of motor dysfunction were discernable in adult *Atoh1^{Cre/+};Vglut2^{fl/fl}* or *Ntsr1^{Cre/+};Vglut2^{fl/fl}* mice. One possible explanation could be that the vesicular subtype switch in granule cells from *Vglut2* to *Vglut1* resulted in a normalization of Purkinje cell firing patterns after the early postnatal period (Fig. 2 vs. Fig. 6), which may, in part, explain the difference in the severity of motor dysfunction between young and adult *Atoh1^{Cre/+};Vglut2^{fl/fl}* mice. Yet, the transporter switch does not explain why *Ntsr1^{Cre/+};Vglut2^{fl/fl}* mice also have discernable motor deficits in the early postnatal period despite not having a primary defect in granule cell development. Another potential explanation for the persistent motor abnormalities in *Atoh1^{Cre/+};Vglut2^{fl/fl}* mice could be the dysfunction of *Atoh1* lineage neurons within the spinal cord (Fig. 4l). However, previous work has shown that elimination of neurotransmission from all (VGluT1 and VGluT2 expression) *Atoh1* lineage neurons within the spinal cord does not cause motor dysfunction[40]. We interpret these data to mean that spinal cord dysfunction is not the main driver of the phenotypes observed in adult *Atoh1^{Cre/+};Vglut2^{fl/fl}* mice. Therefore, in this manipulation, it is likely that other cellular and circuit mechanisms

are the potential sources that promote changes in the severity of motor impairments.

One possible explanation for the differences in severity between developing and adult conditional knockout mice is that the developmental genetic perturbation of *Vglut2* deletion from glutamatergic nuclei neurons can be compensated for by other neurons in the motor circuit. Those cells could be parallel pathways from GABAergic nuclei neurons, as mentioned above, or part of a compensatory mechanism by mature granule and Purkinje cells[21]. In *Ntsr1*[Cre/+];*Vglut2*[fl/fl] mice, these compensatory cells could also include the non-*Ntsr1* expressing glutamatergic cerebellar nuclei neurons, which may also partially explain why their early postnatal motor deficits can be completely recovered by adulthood. Alternatively, the compensation may be initiated by a change in brain-wide networks that control movement during development versus adulthood[63–65]. In neonates, neural activity in the motor cortex resembles that of neurons in the sensory cortex, with neural activity occurring after movements rather than preceding movements[64]. During this period, it is the red nucleus, which receives strong monosynaptic inputs from the cerebellum, that acts as the predominant source of motor output[66]. The motor cortex starts controlling movement after cerebellar function further matures, and a cerebellar-dependent internal model of movement arises in thalamic nuclei that receive dense innervation from the cerebellum and relay information to the motor cortex[63,65]. This cerebellum-involved switch in motor initiation from the red nucleus to the motor cortex may explain why perturbations in cerebellar output pathways cause more severe motor deficits during early postnatal development compared to adulthood. Therefore, a temporally dependent mechanism may ultimately explain why the early postnatal *Atoh1*[Cre/+];*Vglut2*[fl/fl] and *Ntsr1*[Cre/+];*Vglut2*[fl/fl] conditional knockout mice both develop with severely impaired motor reflexes, whereas, in comparison, their adult counterparts present with only mild or completely absent motor dysfunction.

In addition to the difference in the severity of motor deficits between young and old *Atoh1*[Cre/+];*Vglut2*[fl/fl] and *Ntsr1*[Cre/+];*Vglut2*[fl/fl] mice, there is also an intriguing discrepancy between the motor function between the two conditional knockouts during adulthood; adult *Atoh1*[Cre/+];*Vglut2*[fl/fl] mice display mild motor impairments while *Ntsr1*[Cre/+];*Vglut2*[fl/fl] mice exhibit no obvious motor deficits. One potential explanation for this difference could be the additional dysfunction of *Atoh1* lineage neurons within the spinal cord of *Atoh1*[Cre/+];*Vglut2*[fl/fl] mice (Fig. 4l). However, previous work has shown that elimination of neurotransmission from all (VGluT1 and VGluT2 expression) *Atoh1* lineage neurons within the spinal cord does not cause motor dysfunction[40], making this an unlikely driver of the observed motor outcomes. The other difference between *Atoh1*[Cre/+];*Vglut2*[fl/fl] and *Ntsr1*[Cre/+];*Vglut2*[fl/fl] mice is the extent of affected glutamatergic nuclei neurons (Fig. 1 versus Fig. 8). In *Ntsr1*[Cre/+];*Vglut2*[fl/fl] mice, we found that about 50% of glutamatergic cerebellar nuclei neurons are genetically silenced (Fig. 8) while nearly 100% of them are silenced in *Atoh1*[Cre/+]; *Vglut2*[fl/fl] mice (Fig. 1). In both mice during adulthood, only the glutamatergic cerebellar nuclei neurons are affected, and the granule cells of *Atoh1*[Cre/+];*Vglut2*[fl/fl] mice that were silenced during development now appear to function normally as confirmed by in situ histochemistry (Fig. 5) and electrophysiology (Fig. 6). Together, our results suggest that the difference in the recovery of motor function between the two conditional knockouts in adulthood can likely be attributed to compensation mediated by unaffected cells in the motor circuit, especially non-*Ntsr1* expressing glutamatergic cerebellar nuclei neurons. Perhaps some threshold level of glutamatergic cerebellar nuclei neuron function is required to fully recover motor function as shown in *Ntsr1*[Cre/+]; *Vglut2*[fl/fl] mice, which can only be partially recovered by parallel GABAergic nuclei neurons in *Atoh1*[Cre/+];*Vglut2*[fl/fl] mice. It may also be possible that the dysfunction of granule cells in early postnatal developing *Atoh1*[Cre/+];*Vglut2*[fl/fl] mice may have induced long-term changes in motor circuitry that could not be fully resolved in adulthood by compensatory mechanisms, but this is less likely as our data shows that cerebellar cortical function normalizes by this time (Fig. 6). This highlights the importance of glutamatergic cerebellar nuclei neurons in mediating motor function, and the incredible ability of the cerebellum to overcome some, but not all, perturbations of its circuitry. Thus, compensating for abnormal glutamatergic cerebellar nuclei function in diseases with motor symptoms may be key to designing treatments for recovering motor deficits.

### Assessing neural function versus neurogenesis

Our genetic silencing approach provides a compelling in vivo demonstration of how the function of *Atoh1* lineage neurons can be assessed using a functional rather than an anatomical lesion. Previous studies inferred the function of these glutamatergic neurons based on anatomical location and synaptic connectivity[25,67] or tested the function of *Atoh1* lineage neurons using local deletion of the pro-neural *Atoh1* gene, which causes an abnormal or a complete lack of anatomical development of these neurons[29,32,68–70]. These studies have uncovered fundamental and groundbreaking mechanisms for neonatal lethality in *Atoh1* null mice[32,70]. However, we, and others[71], have previously demonstrated that the lack of *Atoh1* lineage neurons can have substantial effects on the development of surrounding neurons[29] that may exacerbate the behavioral effects upon losing *Atoh1* lineage neurons. Instead, our genetic approach is precise (it only affects *Atoh1* lineage or *Ntsr1* expressing neurons) and specific (only the glutamatergic transporter *Vglut2* is deleted). We have used the elimination of vesicular transporters to study the contribution of specific circuit components in several previous studies. In these studies, we did not detect changes in synaptic connectivity or developmental compensation that overcome the functional elimination of neurotransmission[21,33,54,72]. Therefore, we propose that our genetic methods allow for testing how synaptic VGluT2-dependent neurotransmission only in neurons within the intersectional domain influences circuit function and mouse behavior. Our study confirms that *Atoh1* lineage cerebellar nuclei neurons are necessary for the refinement of motor behavior, and we also unveil that *Atoh1* granule cells contribute to social vocalizations during postnatal development.

This study provides in vivo experimental evidence showing that glutamatergic neurons in the cerebellum are critical for the acquisition of motor function and social behaviors. Specifically, the glutamatergic neurons in the cerebellar cortex and nuclei are differentially required for the early postnatal development of motor behavior and social behaviors. These data also raise the possibility that parallel signaling from the GABAergic nuclei neurons may provide some compensation for functional lesions of the glutamatergic nuclei neurons during development, leading to improvements in early motor and social deficits. By leveraging the ability to restore cerebellar-dependent behaviors after normalizing cerebellar cortical function, tapping into cerebellar neurons that communicate social and motor signals to extra-cerebellar regions may provide an ideal therapeutic target to restore motor and non-motor functions after developmental brain injury and disease.

## Methods
### Animals
All mice included in the experiments for this manuscript were housed in a Level 3, AALAS-certified facility. The Institutional Animal Care and Use Committee (IACUC) of Baylor College of Medicine (BCM) reviewed and approved all studies that involved mice. We used the following mice for our experiments: Ai14 (*Rosa*[lsl-tdTomato]; JAX: 007914);[73] *Ai32* (*Rosa*[fsf-ChR2-YFP], JAX: 012569);[74] *Ai65* (*Rosa*[fsf-lsl-tdTomato]; JAX: 021875);[74] *Atoh1*[Cre];[75] *Atoh1*[FlpO] (JAX: 036541);[32] *Ntsr1*[Cre] (MMRRC: 030648);[57] *Vglut2*[IRES-Cre] (JAX: 028863);[76] *Vglut2*[fl] (JAX: 012898)[72]. We used ear clippings from pre-weaned pups for genotyping and identification of

transgenic alleles. Mice of both sexes were used in all experiments. We considered the day the pups were born as postnatal day 0 (P0). We did not observe gross differences in weight between control and *Atoh1^Cre;Vglut2^fl/fl* conditional knockout mice (Adult: control: $N = 7$, weight = $32.2 \pm 2.4$ g; *Atoh1^Cre;Vglut2^fl/fl*: $N = 6$, weight = $28.7 \pm 3.6$ g; $p = 0.413$; P14: control: $N = 4$, weight = $9.75 \pm 0.4$ g; *Atoh1^Cre;Vglut2^fl/fl*: $N = 4$, weight = $8.9 \pm 0.4$ g; $p = 0.161$; P10: control: $N = 5$, weight = $6.2 \pm 0.4$ g; *Atoh1^Cre;Vglut2^fl/fl*: $N = 6$, weight = $5.7 \pm 0.3$ grams; $p = 0.397$). The ages of the adult mice were between two and fourteen months old. All mice were kept under a 14 h/10 h light/dark cycle, a daily temperature between 68 and 72 F, and humidity between 30 and 70%.

## Tissue processing

We collected brain and spinal cord tissue for analyses as previously described[29]. We anesthetized mice with Avertin and tested them for effective anesthesia by toe or tail pinch. Analgesia was also provided. We then accessed the chest cavity and penetrated the heart with a small butterfly needle for perfusion. We first perfused with 1 M phosphate-buffered saline (PBS pH 7.4) to flush out blood until the liver turned clear. Next, we perfused the mouse body with 4% paraformaldehyde (PFA) to fix the tissue. After the tail and hind paws were stiff from fixation, we decapitated the mouse and dissected the brain from the skull or spinal cord from the spinal column. We post-fixed brain and spinal cord tissue in 4% PFA overnight or until the tissue was used for cryoprotection. We cryoprotected the tissue by serial sucrose gradients (15% → 30% sucrose in PBS) at 4 °C, each step until the tissue sinks. Finally, we froze the tissue in an optimal cutting temperature (OCT) solution and stored it at −80 °C. Brain sections were cut into 40 μm free-floating tissue sections, and spinal cord sections were cut into 25 μm sections on the slide. Cut sections were stored at 4 °C until it was used for immunohistochemistry (tissue was stored for a maximum of two months). Tissues from mice expressing the tdTomato allele were stored in aluminum foil at all steps to prevent photobleaching.

## Immunohistochemistry

We stained free-floating tissue sections as previously described[29]. We blocked tissue sections in 500 μL 10% normal goat or normal donkey serum in 0.1% Triton-X in PBS (PBS-T) for 2 h. We then incubated tissue sections overnight at room temperature in 500 μL blocking solution with primary antibodies at desired concentrations. After washing the tissue sections three times in PBS-T, we incubated the tissue for two hours in 500 μL PBS-T with secondary antibodies at desired concentrations. Subsequently, we incubated tissue sections with DAPI (1:500; Sigma-Aldrich; #D9542) in PBS for 10 min. After a final two washes, we mounted the sections using VECTASHIELD Vibrance® mounting medium (Vector Laboratories; #H1700). All mounted slides were stored at 4 °C until they were imaged. We used the following primary antibodies: guinea-pig (gp)-anti-VGluT2 (1:500; Synaptic Systems; #135404), rabbit (rb)-anti-VGluT1 (1:500; Synaptic Systems; #135302), rb-anti-Parvalbumin (PV) (1:1000; Swant; #PV 28), rb-anti-Neurogranin (NG) (1:500; Chemicon; #AB5620), rb-anti-carbonic anhydrase 8 (Car8) (1:500; Proteintech; #12391-1-AP), and sh-anti-tyrosine hydroxylase (TH) (1:500; Millipore; #AB1542). We used the following secondary antibodies which were conjugated to an Alexa-488 fluorophore: goat-anti-gp (1:1000; Invitrogen; #A11073), goat-anti-rb (1:1000; Invitrogen; # A32731), or goat-anti-sh (1:1000; Invitrogen; # A11015).

## In situ hybridization (ISH)

ISH was performed by the RNA ISH Core at Baylor College of Medicine using an automated robotic platform. ISH was used to visualize *Vglut2*, *Vglut1*, *YFP*, and *tdTomato* expression in unfixed, fresh frozen tissue cut in 25 μm-thick coronal brain sections. Digoxigenin (DIG)-labeled mRNA antisense probes against *Vglut2, Vglut1, YFP*, and *tdTomato* were

generated using an RNA DIG-labeling kit from Roche. The specific sequences of the antisense probes that we used were as follows:

*Vglut2:* GGTGCTGGAGAAGAAGCAGGACAACCGAGAGACCATCG AGCTGACAGAGGACGGTAAGCCCCTGGAGGTGCCTGAGAAGAAGGCT CCGCTATGCGACTGCACGTGCTTCGGCCTGCCGCGCCGCTACATCAT AGCCATCATGAGCGGCCTCGGCTTCTGCATATCCTTCGGCATCCGC TGTAACCTGGGCGTGGCCATCGTGGACATGGTCAACAACAGCACTAT CCACCGCGGAGGCAAAGTTATCAAGGAG

*Vglut1:* CAGAGCCGGAGGAGATGAGCGAGGAGAAGTGTGGCTTTG TTGGCCACGACCAGCTGGCTGGCAGTGACGAAAGTGAAATGGAGG ACGAGGCTGAGCCCCCAGGGGCGCCCCCCGCGCCGCCTCCGTCCT ACGGGGCCACACACAGCACAGTGCAGCCTCCGAGGCCCCCGCC CCCTGTCCGGGACTACTGACCACGGGCCTCCCACTGTGGGGCAGTTT CCAGGACTTCCACTCCATACACCTCTAGCCTGAGCGGCAGTGTCG AGGAACCCCACTCCTCCCCTGCCTCAGGCTTAAGATGCAAGTCCTC CCTTGTTCCCAGTGCTGTCCGACCAGCCCTCTTTCCCTCTCAACTGC CTCCTGCGGGGGGTGAAGCTGCACACTAGCAGTTTCAAGGATACCC AGACTCCCCTGAAAGTCGTTCTCCGCTTGTTTCTGCCTGTGTGGGC TCAAATCTCCCCTTTGAGGGCTTTATTTGGAGGGACAGTTCAACCTC TTCCTCTCTTGTGGTTTTGAGGTTTCACCCCTTCCCCCAAGACCCCAG GGATTCTCAGGCTACCCCGAGATTATTCAGGTGGTCCCCTACTCAGA AGACTTCATGGTCGTCCTCTATTAGTTTCAAGGCTCGCCTAACCAATT CTACATTTTTCCAAGCTGGTTTAACCTAACCACCAATGCCGCCGTTC CCAGGACTGATTCTCACCAGCGTTTCTGAGGGA

*YFP:* AGCTGACCCTGAAGTTCATCTGCACCACCGGCAAGCTGCCC GTGCCCTGGCCCACCCTCGTGACCACCCTGACCTACGGCGTGCAGTG CTTCAGCCGCTACCCCGACCACATGAAGCAGCACGACTTCTTCAAG TCCGCCATGCCCGAAGGCTACGTCCAGGAGCGCACCATCTTCTTCAA GGACGACGGCAACTACAAGACCCGCGCCGAGGTGAAGTTCGAGGGC GACACCCTGGTGAACCGCATCGAGCTGAAGGGCATCGACTTCAAGGA GGACGGCAACATCCTGGGGCACAAGCTGGAGTACAACTACAACAGCC ACAACGTCTATATCATGGCCGACAAGCAGAAGAACGGCATCAAGG TGAACTTCAAGATCCGCCACAACATCGAGGACGGCAGCGTGCAGCT CGCCGACCACTACCAGCAGAACACCCCCATCGGCGACGGCCCCGTG CTGCTGCCCGACAACCACTACCTGAGCACCCAGTCCGCCCTGAGCAA AGACCCCAACGAGAAGCGCGATCACATGGTCCTGCTGGAG

*tdTomato:* ATCAAAGAGTTCATGCGCTTCAAGGTGCGCATGGAGG GCTCCATGAACGGCCACGAGTTCGAGATCGAGGGCGAGGGCGAGGG CCGCCCCTACGAGGGCACCCAGACCGCCAAGCTGAAGGTGAC CAAGGGCGGCCCCCTGCCCTTCGCCTGGGACATCCTGTCCCCCCAG TTCATGTACGGCTCCAAGGCGTACGTGAAGCACCCCGCCGACATCC CCGATTACAAGAAGCTGTCCTTCCCCGAGGGCTTCAAGTGGGAG CGCGTGATGAACTTCGAGGACGGCGGTCTGGTGACCGTGACCCAG-GACTCCTCCCTGCAGGACGGCACGCTGATCTACAAGGTGAAGATGCG CGGCACCAACTTCCCCCCCGACGGCCCCGTAATGCAGAAGAAGACCA TGGGCTGGGAGGCCTCCACCGAGCGCCTGTACCCCCGCGACGGCGT GCTGAAGGGCGAGATCCACCAGGCCCTGAAGCTGAAGGACGGCGG CCACTACCTGGTGGAGTTCAAGACCATCTACATGGCCAAGAAGCCCG TGCAACTGCCCGGCTACTACTACGTGGACACCAAGCTGGACATCAC CTCCCACAACGAGGACTACACCATCGTGGAA

## Nissl staining

Nissl staining was performed with the tissue sections mounted and dried overnight on the slides. The next day, the mounted sections were immersed in 100% xylene two times for five minutes each time. Subsequently, they were placed through a rehydration series of 3 immersions in 100% ethanol, followed by 95% ethanol, 70% ethanol, and tap water, with each step lasting 2 min. Afterward, the sections were immersed in cresyl violet solution for approximately two minutes. They were then dehydrated following a reversed order of the rehydration series and followed by a final immersion in xylene, with each step lasting up to one minute. Coverslips were mounted on the slides immediately after using Cytoseal XYL mounting media (Thermo Scientific, Waltham, MA, USA, #22-050-262).

## Microscopy and image processing

We acquired photomicrographs of cerebellar sections using a Leica DM4000 B LED microscope with a DPX365FX camera or using a Zeiss Axio Imager.M2 microscope with an AxioCam MRc5 camera. We stitched the whole mount images for the intersectional lineage tracing together using the Adobe Photoshop Photomerge function. We adjusted color brightness and balance using ImageJ software, and we cropped the images to the desired size for figures using Adobe Illustrator.

## In vivo electrophysiology

We performed in vivo electrophysiological recordings of Purkinje cells according to previously described publications[29,77]. Prior to surgery, mice were anesthetized using a mixture of ketamine (80 mg/kg) and dexmedetomidine (16 mg/kg). During the surgery, the mouse was placed on a heated surgery pad and received additional isoflurane (1%) when necessary. The mouse's head was stabilized by ear bars in a stereotaxic surgery rig. When the mouse was fully anesthetized, we removed the hair from the head and made an incision in the skin over the anterior part of the skull. Next, we used a dental drill to make a large craniotomy over the cerebellar cortex. The craniotomy spanned from around bregma −5.60 mm to −6.64 mm, and from around 0.5–3 mm lateral to bregma. Purkinje cells across several lobules were randomly sampled from this craniotomy, preventing any potential bias from local activity patterns (Supplementary Fig. 5). We recorded neural activity using tungsten electrodes and digitized the signals into Spike2 (CED). Purkinje cells were identified based on their depth within the cerebellum (0–2 mm from the brain surface) and the presence of clear complex spikes. We used both male and female mice for our recordings.

## Analysis of in vivo electrophysiology

All electrophysiological recordings were spike sorted in Spike2. We separately sorted out Purkinje cell simple spikes and complex spikes. Complex spikes were recognized by their large action potential waveform that is followed by 3–5 smaller spikelets. We only included in our final analyses Purkinje cell recordings with clearly identifiable complex spikes, a good signal-to-noise ratio, and a minimal recording duration of 60 s[78]. For this study, we defined "firing rate" as the average number of spikes per second. We defined "CV" as the standard deviation of the ISI divided by the mean of the ISI. We defined "CV2" as the mean of the difference between all adjacent ISI divided by the sum of all adjacent ISI.

## Behavioral analyses—breathing assay

We tested respiratory parameters in room air as described in our previous publication[32]. In short, we placed mice in plethysmography chambers to acclimate for one hour. After the habituation period, we used Phonemah software (DSI) to acquire breath waveforms and used custom-written MATLAB (Mathworks) code to derive Tidal Volume, Respiratory Frequency, and Minute Volume parameters.

## Behavioral analysis—negative geotaxis

We tested the negative geotaxis reflex at P7, P9, and P11 as previously described and shown[47]. Mice were placed head down on a negative incline (−35°) that was covered with a sterile Poly-Lined drape. We measured the time until mice turned 90° in either direction. Mice were tested three times at each time point. We suspended the test if mice did not turn within 60 s or fell down the slope. We recorded the falls that occurred within 60 s.

## Behavioral analysis—surface righting reflex

We tested the surface righting reflex at P7, P9, and P11, as previously described and shown[47]. Mice were placed in the supine position in an empty, clean cage. We measured the time until the mice turned onto their four paws. Mice were tested three times at each time point. We suspended the test if mice did not turn within 1 min (60 s).

## Behavioral analysis—ultrasonic vocalizations

We measured pup ultrasonic vocalizations in a social isolation task at P7, P9, and P11. The ultrasonic vocalizations of each animal were monitored for 2 min in a sound-attenuating chamber (Med Associates Inc.). For *Atoh1^Cre/+;Vglut2^fl/fl* mice and littermate controls, vocalizations were recorded using a CM16 microphone (Avisoft Bioacoustics). Sound was amplified and digitized using UltraSoundGate 416H at a 250 kHz sampling rate, and bit depth of 16 while Avisoft RECORDER software was used to collect the recordings. For *Ntsr1^Cre;Vglut2^fl/fl* mice and littermate controls, we measured vocalizations using a Noldus microphone and UltraVox XT software. Due to minimal congruency between the number of vocalizations recorded between these two recording systems[79], we only compared mutants to control littermates whose vocalizations were measured using the same system.

## Behavioral analysis—open field assay

We measured open-field activity using automated Fusion software. Mice were placed in the center of an open field (40x40x30 cm chamber) that has photo beams for detecting horizontal and vertical activity. The chamber was placed in a room with the light set to 200 lux and ambient white noise to 60 dB. Each mouse was allowed to explore the chamber for 30 min.

## Behavioral analysis—3-chamber assay

We used the three-chamber test[80,81] to assess the sociability of adult mice. The apparatus consisted of three chambers—a center chamber with doorways to two side chambers, all of equal dimensions (42.5 cm length; 17.5 cm, width; 23 cm height). For three days prior to testing, the age- and sex-matched mice to be used as partners for *Atoh1^Cre/+;Vglut2^fl/fl*, *Ntsr1^Cre;Vglut2^fl/fl*, and their control littermates during the social interaction test were placed under a wire cup for 1 h each day. The assay consisted of a 10-min habituation session followed by a 10-min test phase performed in dim lighting conditions (15 lux). During the habituation session, the test subject was placed in the center chamber and allowed to freely explore for 10 min. Once the subject returned to the center chamber, the doorways to the side chambers were blocked with plexiglass walls. Before the test phase starts, a novel partner mouse (pre-habituated to the apparatus 3 days prior) is placed under a wire cup in one side chamber, while a novel object (Lego block of similar size and color) is placed under a wire cup in the other side chamber. The plexiglass walls covering the doorways are then lifted, and the test session begins as the test subject is allowed to freely investigate all three chambers for 10 min. The amount of time spent interacting with either the novel object or novel mouse during the two phases was scored manually by an experimenter blinded to genotype during the assay using ANY-Maze. Additional activity data that was acquired from each chamber during the two sessions was calculated automatically using ANY-Maze. The placement of the novel mouse and novel object within the side chambers was randomized to prevent chamber bias.

## Behavioral analysis—rotarod

We assessed rotarod performance using a previously established protocol[53]. As is standard for this task, mice were placed on an accelerating rod on which they locomote according to the motion of the rod (4–40 rpm over 5 min) (ENV-576M and ENV-571M, Med Associates, Inc.). Time was stopped and noted when one of three events occurred; the mouse fell off, the mouse made two consecutive rotations while hanging on the rod without walking, or the mouse successfully stayed on the rod for the total duration of the trial (5 min). We recorded the trial duration for four trials per day for three consecutive days.

## Behavioral analysis–tremor

We recorded tremors using our custom-built tremor monitor[51]. The tremor monitor consists of a lightweight plastic container that is suspended in the air by eight elastic cords, one on each corner. The container is large enough for mice to move around and explore freely, providing us with readings related to movement. An accelerometer is mounted onto the underside of the container, where it functions to convert the detected movements into an electrical signal that is digitized using a Brownlee amplifier and records using Spike2 software. We also used the Spike2 software to detect the tremor power using a Fourier transform analysis based on two full minutes of mouse movements. For the recordings, mice were placed in the tremor monitor and were allowed to acclimate for three minutes. After this period, the recordings for analysis were initiated once the animals started actively exploring and utilizing the space.

## Statistical analysis

All statistical analyses were performed using MATLAB (Mathworks). When performing a two-way (sex*genotype) ANOVA, we did not observe an effect of sex or an interaction effect between sex and genotype on pup behavior. We, therefore, combined mice from both sexes in the experiments and statistical analyses. We performed a two-tailed T-test to assess differences between control and conditional knockout mice when only a one-time point was involved. For analyses of pup behavior and rotarod performance, we performed a repeated measures ANOVA analysis with a Tukey post hoc analysis to test for differences at each time point. For the analyses of electrophysiological recordings in Purkinje cells, we used a linear mixed model analysis with genotype as a fixed variable and mouse number as a random variable. We calculated Cohen's $d$ for effect size (and reported these in the figure legends) by dividing the absolute difference in the parameter average of each group by the combined standard deviation.

## Reporting summary

Further information on research design is available in the Nature Portfolio Reporting Summary linked to this article.

## Data availability

The data that support the findings of this study are available from the corresponding author. The data generated in this study are available in the source data provided in this paper. Source data are provided in this paper.

## Code availability

The code used for this study is available from the corresponding author.

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

## Acknowledgements

This work was supported by Baylor College of Medicine (BCM), Texas Children's Hospital, The Hamill Foundation, and the National Institutes of Neurological Disorders and Stroke (NINDS) R01NS100874, R01NS119301, and R01NS127435 to R.V.S. MEvdH was supported by a postdoctoral award from the Dystonia Medical Research Foundation (DMRF) and NINDS (1K99NS130463-01). L.H.K. was supported by a postdoctoral award from the DMRF. Research reported in this publication was supported by the Eunice Kennedy Shriver National Institute of Child Health & Human Development of the National Institutes of Health under Award Number P50HD103555 for the use of the Cell and Tissue Pathogenesis Core and In Situ Hybridization Core (the BCM IDDRC). The content is solely the responsibility of the authors and does not necessarily represent the official views of the National Institutes of Health. We thank Yuan Chang and Dr. Russell Ray (BCM) for kindly sharing the primer sequences that specifically detect the *Vglut2* mRNA sequence that is flanked by the *LoxP* sites.

## Author contributions

Conceptualization: M.Evd.H., A.G.R.H., and R.V.S. Methodology: M.Evd.H., A.G.R.H., L.H.K., T.L., and R.V.S. Software: M.Evd.H. Validation: M.Evd.H. and A.G.R.H. Formal analyses: M.Evd.H. and A.G.R.H. Investigation: M.Evd.H., A.G.R.H., L.H.K., D.J.K., R.M.P., and R.V.S. Data curation: M.Evd.H., A.G.R.H., L.H.K., D.J.K., R.M.P., and T.L. Writing—original draft: M.Evd.H., A.G.R.H., and R.V.S. Writing—review & editing: M.Evd.H., A.G.R.H., L.H.K., D.J.K., R.M.P., T.L., and R.V.S. Visualization: M.Evd.H., A.G.R.H., and L.H.K. Supervision: R.V.S. Project administration: M.Evd.H., A.G.R.H., and R.V.S. Funding acquisition: M.Evd.H., L.H.K., and R.V.S.

## Competing interests

The authors declare no competing interests.
