## [Peer Review File · Nature Communications]

Glutamatergic Cerebellar Neurons Differentially Contribute to the Acquisition of Motor and Social BehaviorsREVIEWER COMMENTS

Reviewer #1 (Remarks to the Author):

Increasing evidence indicate that in addition to motor control the cerebellar cortex and nuclei regulate social behaviors. In a series of carefully designed experiments, van der Heijden et al examine the consequences of VGluA2 deletion from Atoh1 lineage neurons and Ntsr1-expressing neurons in the regulation of motor, non-motor behaviors and cerebellar physiology in early postnatal (P7-P11) and adult mice. The authors find important roles for Vglut2-dependent transmission within the cerebellar cortex and DCN in the acquisition of motor and social behaviors. These data are very interesting and highlight the importance to take into account the developmental dimension. However, despite my excitement about the novel insights explored in the present study, I have several suggestions that could improve the manuscript and strengthen the conclusions.

Comments

- Fig 1: The quality of the images provided is remarkable. However, basic characterization of the mutant mice is missing. For instance, the general morphology of the cerebellum should be included (please provide coronal and sagittal sections at low magnification). The analysis of the cerebellar morphology across lobules and the cerebellar zonal expression should be explored. Moreover, the potential impact of Vglut2 deletion on the distribution of other cerebellar cell types (Interneurons, granules cells or glial cells) should be analyzed. Finally, the cerebellar expression/distribution of Slc17a7/VGluA1 at P7 and in adult should be provided. Please address all these points in the results.

- Fig 2: Motor and non-motor behaviors are strongly affected in mutant mice at early postnatal stage. The authors link these effects to the invalidation of the vglut2 in the cerebellum. However, Atoh1, as VGlu2, is highly expressed in the dorsal spinal cord. Do the authors determine whether the deletion of VGluA2 occurred in the dorsal SC in the mutant mice? Please address this point. If so, interpretation of the results related to motor behaviors should be reconsidered and additional experiments should be performed to rule out a contribution of altered SC function.

- Fig 3: What is missing in Figure 3 is where the recordings have been performed. Recording should be performed in lobules IV and V most required for motor function and compared to recording done in Crus I/II lobules more involved in the control of social behaviors. Please address these points in the results. Please also specify whether the recording have been performed in males and/or females.

- Fig 4: The intersectional lineage approach is extremely elegant and again the images provided are of high quality. Looking at carefully panel E, I am not sure that what the authors identified as VTA is indeed VTA. Please perform a double staining with an antibody against TH to confirm that the projections surround TH/DAT neurons.

- Fig 5: As mentioned by the authors the Ntsr1-Cre mouse line also drives the recombination in cortico-thalamic neurons, however as in the cerebellum Slc17a6, the gene encoding VGluA2 is transiently

expressed in the cortex during the development (He et al., 2012 PMID 23136427). Therefore, how rule out that the motor phenotype do not rely on altered cortico-thalamic functions rather than the lack of VgluA2 in the DCN? Please address this issue in the results. Based on their interpretation, it looks like the authors assume that in the DCN Atoh1/VGluA2 neurons and Nstr1/VGluA2 neurons are the same. What is the degree of overlap between Atoh1 and Nstr1? smFISH should help to address this issue. If they partially overlap the interpretation of the results and the discussion should be modified accordingly.

- Fig 6: See my comment above. Please Indicate in which lobules (vermis vs hemisphere) the recording have been performed.

- Fig 7: Please consider comments related to Fig 2. If altered SC functions are confirmed, this could account for the maintenance of the motor phenotype. It will be also important to show whether the motor phenotype is present in adult mice lacking VGlu2A in Nstr1 neurons. ? Please address this issue in the results

Reviewer #2 (Remarks to the Author):

Summary

The authors use cell-specific deletion of VGluT2 which likely eliminates vesicular release from excitatory synapses, to dissect changes arising from granule cell input and CN output. They first looked at developing Atoh VGluT2 knock-out mice, which show deficits in both cell types, granule cells and nuclear cells in young developing mice. They then compare this to Ntsr1 VGluT2 knock-out mice to isolate the effects of nuclear output from granule cell output in development. They observe a range of behavioral changes that vary over ages. At these young age, motor assays are righting reflex and negative geotaxis, plus vocalization are abnormal in the Atoh VGluT2 knock-out mice mice. They also see a somewhat subtle changes in simple spike firing in anesthetized mice. However, there appear to be motor deficits but only mild social deficits in the Ntsr1 VGluT2 knock-out mice, although they say there are no social deficits.

They then look at adult Atoh VGluT2 knock-out mice because there is a developmental switch in the cerebellum by which the granule cells switch from using VGluT2 to VGluT1 in the adult cerebellum. They feel that this allows them to differentiate the contribution of nuclear and granule cell outputs, and show that normal granule cell but impaired nuclear cell function leads to motor only deficits in the adult.

Strengths of this manuscript is that the authors strengthen the evidence that these early motor assays have a cerebellar component, which is important. I also appreciate that they show male and female data separately, this should be standard for the field.

Major Points

While the study is interesting, there are a number of caveats that weaken the strength of their findings and mean that the interpretation of these results is not entirely clear.

1. The authors refer to the Atoh VGlut2 mice as being conditionally targeted to developing synapses, due to the developmental switch to VGlut1 in granule cells. But they do not demonstrate that this actually occurs. There is good evidence that compensatory mechanisms exist during development, so it is possible that normal development does of these synapses not occur when VGlut2 is deleted, and/or that compensatory changes exist at these synapses during development. In my opinion, it would be essential for the authors to demonstrate that the granule cell connections are non-functional but existing in the developing mouse, and that they look and function normally in the adult mouse. Directly assaying synapses using electrophysiology would be a good way to demonstrate this, combined with more morphological analysis and immunohistochemistry (showing normal VGlut1 in the adult, for example is required).

2. Unfortunately, the same caveats exist for the nuclear synapses as for the granule cell synapses – do they develop normally and are they sure that there is no compensatory changes occurring that make the synapses functional even with the VGlut2 deletion? This would be much harder to address experimentally, so it might be best to dial back their claims. The major referential support for the use of Ntsr1-Cre mice to target excitatory CN neurons is a preprint (citation #30) produced by the authors of the current study. Since that work has not undergone peer review, the authors should provide stronger rationale and evidence for the use of Ntsr1 VGlut2 mice to target CN neurons. In the preprint, they describe that there are few cells labeled in the fastigial nucleus in the Ntsr1-Cre mice. It would be nice to show this labeling in this paper, since it is important for understanding how much of the nuclear output is being labeled. Their statement from the pre-print seems to weaken their findings, since if they are missing a large population of excitatory outputs from the nuclei, it is hard to conclude that the nucleus is not contributing to social output. Furthermore, the authors state that there are no changes in social behavior in development in Ntsr1 VGlut2 mice, but in fact, do not discuss the apparent delay in the vocalization that they appear to observe in these mice in development, since they show that one measure is significantly different at their oldest developmental age (P11). Again, this seems to weaken the strength of their claims. They could dial back the strength of their claims, but it would also be great if they included further social behaviors, especially in the adult.

3. An opportunity to strengthen their findings has been missed by not comparing results from their two mice strains in the adult, since the Ntsr1 VGlut2 mice have not been examined in the adult. If they observed similar results in the Ntsr1 VGlut2 mice as observed in the Atoh VGlut2 adult mice, this would help strengthen their findings, and might in part address the questions raised by the limited Ntsr1 expression.

4. The author discuss the Atoh VGlut2 knock-out mice as being “rescued” in the adult. A rescue implies that the authors are doing some intervention, but they have not, they are making this claim based on

normal development, which (as described in point 1) they have not actually shown. Even if they show that adult granule cell synapses are completely normal, I think that “rescue” is not the appropriate term here, it feels misleading. Please rewrite.

Minor Points

1. Typo line 90 “in a social isolation paradigm during.” During what?
2. The authors are encouraged to use different stars to show different degrees of significance. Some of their findings are just significant, whereas others are very robust, but they are all shown with 1-star significance, which makes it harder to interpret, and in fact, probably weakens their stronger findings in the eyes of the reader.
3. The way that the authors represent the 3-chamber test results is not standard, and it would be nice to see more data here, since there could be subtle differences that are not detected with the data presented this way. Since this is a big part of their claim, further social tests would help strengthen these findings.
4. The authors say that there is labeling in layer 5 in the cortex in the Ntsr1 mice, and it would be useful to demonstrate if those cells express VGluT2, and would therefore be affected in these mice.
5. Please don't use the same abbreviation for the cochlear nucleus as for the cerebellar nuclei in other figures (CN).
6. Page 4 line 27: Please add a citation for claim that CFs are the only source of VGluT2 from the Atoh1 lineage at this age.
7. The title seems to imply that the authors are examining different CN neurons, rather than comparing CN neurons to other cerebellar neurons (granule cells). Please revise.

Reviewer #3 (Remarks to the Author):

The manuscript by van der Heijden and colleagues uses Cre-flox recombination in mutant mice to delete VGluT2-mediated glutamatergic neurotransmission in Atoh1+ and Ntsr1+ cell lineages. Central to the manuscript is the finding that VGluT2-mediated neurotransmission from Atoh1+ cell lineages disrupts motor reflexes and social vocalizations in juvenile mice compared to the motor only effects of the more restricted deletion of Vglut2 in Ntsr1+ cell lineages. Based on these differences, the authors claim VGluT2 neurotransmission from cerebellar cortical versus (DCN) nuclei neurons differentially controls the acquisition of motor and social behaviors. Unfortunately this claim is not sufficiently supported by the data. The authors need to provide more convincing data supporting the specificity of granule cell versus DCN neuron deletion of VGluT2-mediated neurotransmission in mice. An additional concern are the conclusions made about the comparative neurocircuitry of Atoh1Cre/+;Vglut2fl/fl and control mice based on electrophysiological data with low sampling power.

Results line 167. “substantiating a role for glutamatergic cerebellar neurons in shaping these behaviors.” As it is not possible to separate the effects of ablating VGLUT2-mediated neurotransmission in granule cell and DCN lineages from that of extra-cerebellar glutamatergic Atoh+ lineages in Figures 2 and 3 data, the data does not substantiate the role of these cerebellar cell types and this claim and should be removed.

Line 173. “mediated by abnormal cortical function” should be edited to “associated with abnormal cortical function”

Supp Figure 1 title typo: “in” should be “is”. Histograms lack +/- SEM bars.

Figure 2 and Figures 3D-I, 5B-D, 6D-I. Text should be added to the legend to indicate the larger open circles each represent the mean for the male or female cohorts for the control and mutant strains at each age; a horizontal bar rather than an open circle to denote the mean would be an improvement. It is also not clear what the shaded areas represent. It looks like a way of plotting the distribution of the data, but the authors should say so if that's the case. It would be a little easier to read if different symbols (as in Fig. 7) or colors were used to represent individual data points for male vs. female. It's true that males and females are in different columns but they're so close together they're hard to distinguish. Clearer annotation of the group means and individual data points to distinguish between groups (males vs females, cells from the same mouse) applies to figure 2 but also figures 3D-I, 5B-D, 6D-I.

Figure 3 and Figure 6 data. The number of animals and cells sampled (mouse n = 3, cells n 12-15) is unusually low. The low sampling power of Figure 3 and 6 data reduces the chance of detecting a true effect in Atoh1Cre/+; Vglut2fl/fl mutants when compared to controls. As such the claims that “Vglut2 loss from the Atoh1 lineage resulted in....no change in simple spike pattern or regularity (Fig. 3E-F)....did not observe any differences in complex spike frequency, pattern, or regularity (Fig. 3H-I)” (lines 183-186) are not sufficiently tested. Similarly, insufficient sampling questions the findings of no difference in simple and complex spike firing in adult Atoh1Cre/+; Vglut2fl/fl mutants compared to controls. (Figure 6).

Figure 3. Text should be added to the legend to indicate CV is Purkinje cell spike pattern and CV2 Purkinje cell spike regularity

Figure 4 – Supplement 1. No mention is made of the arrow annotations in the legend.

Figure 5. Restricted ablation of VGLUT2-mediated neurotransmission in DCN neurons is predicated on Ntsr1-Cre recombinase-mediated deletion. However, other than a reference to Allen Brain Atlas data, no data on Ntsr1 expression in the DCN is provided and it is not clear to what extent an Ntsr1-Cre; VGLUT2fl/fl background would deplete Vglut2 in the DCN. Indeed no data is included demonstrating how effective and expansive this approach is in deleting Vglut2 in the DCN. At a minimum, the authors should provide comparative VGLUT2 expression data in the DCN Ntsr1-Cre; VGLUT2fl/fl mice and controls. Ideally this would be supported by the inclusion of detailed Ntsr1-Cre fate-mapping in the DCN to identify the

targeted nuclei.

Figure 7A-C. Tremor effects are mild. How does the magnitude of the decrease in tremor activity in Atoh1Cre/+; VGluT2fl/fl mice compare to the change observed with deleting GABAergic neurotransmission from Purkinje cells?

Figure 7. For consistency, the authors should annotate the means for female and males.

Results text (lines 242-247) discussing the possibility social deficits recorded in the early postnatal Atoh1Cre/+;Vglut2fl/fl mice are caused by abnormal function within the cerebellar cortex are speculative and better placed in the discussion section.

The closing results section statement (lines 296-298) is not sufficiently supported by the data – see Fig. 5 concerns with the lack of data supporting VGluT2 deletion in Ntsr1-Cre; VGluT2fl/fl mice: “Together, our findings indicate that altering fast neurotransmission from glutamatergic cerebellar nuclei neurons in the developing cerebellar circuit later obstructs the proper execution of motor functions but does not impair social behaviors in adult mice.”

Discussion. Similar to the results sections, the statements “genetic elimination of neurotransmission from glutamatergic nuclei neurons did not impair the acquisition of social vocalizations in early postnatal mice” and “We conclude that intact VGluT2-mediated neurotransmission from glutamatergic nuclei neurons in postnatal mice is essential for the acquisition and coordination of movements, but it is not required for the acquisition and maintenance of social behaviors that require intact cerebellar function” are not sufficiently supported by the data.

Are there weight differences between Atoh1Cre/+; Vglut2fl/fl (or Ntsr1-Cre; Vglut2fl/fl) and control pups at the ages tested for motor and vocal activity during development? The inclusion of weight data would assuage a concern differences are confounded by developmental delay in the mutants.

Statistical analysis. Are all data equally balanced for sex? If so, please state. If not and the data is separated by sex in figures, the n of each sex per group should be included in the legends.

REVIEWER COMMENTS

We would like to thank all three reviewers for providing excellent suggestions that have enabled us to strengthen our manuscript and enhance the impact of the findings. We have addressed each of the comments by altering the text, providing additional data, and revising the figures as requested. Based on the reviewers' outstanding suggestions and comments, we have included 3 additional main figures and 5 additional supplemental figures in our revised manuscript.

Below are our explanations for how we have altered the manuscript in this revised version. The Reviewer's comments are written in black, and our responses are written in blue.

Reviewer #1 (Remarks to the Author):

Increasing evidence indicate that in addition to motor control the cerebellar cortex and nuclei regulate social behaviors. In a series of carefully designed experiments, van der Heijden et al examine the consequences of VGluA2 deletion from *Atoh1* lineage neurons and *Ntsr1*-expressing neurons in the regulation of motor, non-motor behaviors and cerebellar physiology in early postnatal (P7-P11) and adult mice. The authors find important roles for Vglut2-dependent transmission within the cerebellar cortex and DCN in the acquisition of motor and social behaviors. These data are very interesting and highlight the importance to take into account the developmental dimension. However, despite my excitement about the novel insights explored in the present study, I have several suggestions that could improve the manuscript and strengthen the conclusions.

Comments

- Fig 1: The quality of the images provided is remarkable. However, basic characterization of the mutant mice is missing. For instance, the general morphology of the cerebellum should be included (please provide coronal and sagittal sections at low magnification). The analysis of the cerebellar morphology across lobules and the cerebellar zonal expression should be explored. Moreover, the potential impact of Vglut2 deletion on the distribution of other cerebellar cell types (Interneurons, granules cells or glial cells) should be analyzed. Finally, the cerebellar expression/distribution of *Slc17a7*/*VGluA1* at P7 and in adult should be provided. Please address all these points in the results.

We thank the reviewer for remarking on the quality of our images.

In our revised manuscript, we have included 3 additional supplemental figures to characterize the cerebella of *Atoh1*^{Cre/+};*Vglut2*^{fl/fl} conditional knockout mice. We did not find differences in the general morphology between control and conditional knockout mice (Supplemental figure 3), we did not find changes in cerebellar zonal expression as visualized by *VGluT2*⁺ distribution in anterior and posterior cerebellar zones (Supplemental figure 2), and we did not find changes in interneurons or Purkinje cells (Supplemental figure 4).

We also included one additional main figure that shows the expression of *VGluT1* at P7 and in adult mice (Figure 5), that similarly shows no differences between control and conditional knockout mice.

- Fig 2: Motor and non-motor behaviors are strongly affected in mutant mice at early postnatal stage. The authors link these effects to the invalidation of the *vglut2* in the cerebellum. However, *Atoh1*, as *VGlu2*, is highly expressed in the dorsal spinal cord. Do the authors determine whether the deletion of *VGluA2* occurred in the dorsal SC in the mutant mice? Please address this point.

If so, interpretation of the results related to motor behaviors should be reconsidered and additional experiments should be performed to rule out a contribution of altered SC function.

We have now discussed the contribution of altered SC function in our discussion:

“Another potential explanation between the discrepancy in motor deficits in adult *Atoh1*^{Cre/+};*Vglut2*^{fl/fl} and *Ntsr1*^{Cre/+};*Vglut2*^{fl/fl} mice could be a dysfunction of *Atoh1* lineage neurons within the spinal cord (Fig. 4I). However, previous work has shown that elimination of neurotransmission from all (VGluT1 and VGluT2 expression) *Atoh1* lineage neurons within the spinal cord does not cause motor dysfunction.³⁵ We believe the spinal cord dysfunction is not the main driver of the phenotypes observed in adult *Atoh1*^{Cre/+};*Vglut2*^{fl/fl} mice.”

- Fig 3: What is missing in Figure 3 is where the recordings have been performed. Recording should be performed in lobules IV and V most required for motor function and compared to recording done in Crus I/II lobules more involved in the control of social behaviors. Please address these points in the results. Please also specify whether the recording have been performed in males and/or females.

This is a great point as several previous studies have suggested that lobules IV/V and crus I/II are differentially important for motor and social behaviors. Our genetic manipulation affects granule cells across the cerebellum, in multiple regions. In the revised manuscript, we have included additional immunofluorescent images of VGluT2 and VGluT1 expression in lobules V and crus I in Figures 1 and 5. These confirm that our genetic manipulation affects neurons across the cerebellar cortex.

Our recordings are targeted through a broad area of the cerebellar cortex and not targeted to specific regions, yielding a wide sampling of neurons from multiples lobules and cerebellar areas, including but not limited to lobules IV/V and Crus I. To visualize this, we have included sample histology from one of our recorded mice (Supplemental figure 5) and included additional detail in our methods section. To strengthen the confidence in our initial findings and further reduce potential bias from region-specific recordings, we have also performed additional recordings.

We have included a statement in the methods confirming that we used both male and female mice for our recordings.

- Fig 4: The intersectional lineage approach is extremely elegant and again the images provided are of high quality. Looking at carefully panel E, I am not sure that what the authors identified as VTA is indeed VTA. Please perform a double staining with an antibody against TH to confirm that the projections surround TH/DAT neurons.

Thank you for appreciating the elegance of our experimental design and the quality of images. We no longer have access to the triple-transgenic mice used to generate figure 4, and re-obtaining these mice, by regenerating triple-transgenic mice from scratch would prevent a timely resubmission of our manuscript. However, please see below for several ways that we were immediately able to provide some clarity on this excellent point.

The identification of VTA in figure 4 was based on the Allen Brain Atlas. Confirmation of direct excitatory projections from the cerebellar nuclei to the VTA, with double staining for TH/DAT, has been published in two independent previous papers (Judd et al., 2021 and Carta et al., 2019). We also included a high magnification of tdTomato⁺ projections from *Ntsr1*^{Cre} expressing nuclei neurons intermingled with TH⁺ VTA neurons.

- Fig 5: As mentioned by the authors the *Ntsr1*-Cre mouse line also drives the recombination in cortico-thalamic neurons, however as in the cerebellum *Slc17a6*, the gene encoding VGluA2 is transiently expressed in the cortex during the development (He et al., 2012 PMID 23136427). Therefore, how rule out that the motor phenotype do not rely on altered cortico-thalamic functions rather than the lack of VgluA2 in the DCN? Please address this issue in the results. Based on their interpretation, it looks like the authors assume that in the DCN *Atoh1*/VGluA2 neurons and *Ntsr1*/VGluA2 neurons are the same. What is the degree of overlap between *Atoh1* and *Ntsr1*? smFISH should help to address this issue. If they partially overlap the interpretation of the results and the discussion should be modified accordingly.

This is a good point that was raised by multiple reviewers. We have included a new figure, figure 8, to show overlap between *Ntsr1^{Cre}* and *Vglut2* using a double probe FISH, for *Ntsr1^{Cre}*-driven YFP expression and *Vglut2*. We find that the co-expression of YFP and *Vglut2* is only present in the cerebellar nuclei. We have further quantified the YFP and *Vglut2* co-expression. We found a partial overlap and have interpreted the results and modified the discussion accordingly.

- Fig 6: See my comment above. Please Indicate in which lobules (vermis vs hemisphere) the recording have been performed.

A related concern was raised above. Please see our responses and comments above.

- Fig 7: Please consider comments related to Fig 2. If altered SC functions are confirmed, this could account for the maintenance of the motor phenotype. It will be also important to show whether the motor phenotype is present in adult mice lacking VGlu2A in *Ntsr1* neurons. ? Please address this issue in the results

We have now discussed the contribution of altered SC function in our discussion:

“One potential explanation for this difference could be the additional dysfunction of *Atoh1* lineage neurons within the spinal cord of *Atoh1^{Cre/+};Vglut2^{fl/fl}* mice (Fig. 4I). However, previous work has shown that elimination of neurotransmission from all (VGluT1 and VGluT2 expression) *Atoh1* lineage neurons within the spinal cord does not cause motor dysfunction,⁴⁰ making this an unlikely driver of the observed motor outcomes.”

Reviewer #2 (Remarks to the Author):

Summary

The authors use cell-specific deletion of VGluT2 which likely eliminates vesicular release from excitatory synapses, to dissect changes arising from granule cell input and CN output. They first looked at developing *Atoh* VGluT2 knock-out mice, which show deficits in both cell types, granule cells and nuclear cells in young developing mice. They then compare this to *Ntsr1* VGluT2 knock-out mice to isolate the effects of nuclear output from granule cell output in development. They observe a range of behavioral changes that vary over ages. At these young age, motor assays are righting reflex and negative geotaxis, plus vocalization are abnormal in the *Atoh* VGluT2 knock-out mice. They also see a somewhat subtle changes in simple spike firing in anesthetized mice. However, there appear to be motor deficits but only mild social deficits in the *Ntsr1* VGluT2 knock-out mice, although they say there are no social deficits.

They then look at adult Atoh VGLuT2 knock-out mice because there is a developmental switch in the cerebellum by which the granule cells switch from using VGLuT2 to VGLuT1 in the adult cerebellum. They feel that this allows them to differentiate the contribution of nuclear and granule cell outputs, and show that normal granule cell but impaired nuclear cell function leads to motor only deficits in the adult.

Strengths of this manuscript is that the authors strengthen the evidence that these early motor assays have a cerebellar component, which is important. I also appreciate that they show male and female data separately, this should be standard for the field.

Thank you for these kind comments. We appreciate you noticing our efforts in experimental design and rigor.

Major Points

While the study is interesting, there are a number of caveats that weaken the strength of their findings and mean that the interpretation of these results is not entirely clear.

1. The authors refer to the Atoh VGLuT2 mice as being conditionally targeted to developing synapses, due to the developmental switch to VGLuT1 in granule cells. But they do not demonstrate that this actually occurs. There is good evidence that compensatory mechanisms exist during development, so it is possible that normal development does of these synapses not occur when VGLuT2 is deleted, and/or that compensatory changes exist at these synapses during development. In my opinion, it would be essential for the authors to demonstrate that the granule cell connections are non-functional but existing in the developing mouse, and that they look and function normally in the adult mouse. Directly assaying synapses using electrophysiology would be a good way to demonstrate this, combined with more morphological analysis and immunohistochemistry (showing normal VGLuT1 in the adult, for example is required).

The reviewer raises several excellent points in this comment. We have now included a new figure, figure 5, that demonstrates the expression of VGLuT1 in P7 and adult mice. We do not observe any differences between control and conditional knockout mice, confirming that we do not see a developmental effect in synapse compensation.

We have also used *in vivo* electrophysiology to observe the neuronal activity downstream from the granule cells, and the Purkinje cells. We find that these neurons have abnormal spiking activity at P7, but not in adulthood. Together with the anatomical data, we surmise that this is reflective of the change in granule cell neurotransmission.

We have used the elimination of vesicular transporters to study the contribution of specific circuit components in several previous studies. In these studies, we have never found changes in synaptic connectivity or developmental compensation that overcomes the functional elimination of neurotransmission. We have now included this statement with citations in our discussion in the paragraph under “Assessing neural function versus neurogenesis”.

2. Unfortunately, the same caveats exist for the nuclear synapses as for the granule cell synapses – do they develop normally and are they sure that there is no compensatory changes occurring

that make the synapses functional even with the VGlut2 deletion? This would be much harder to address experimentally, so it might be best to dial back their claims.

Please see our response to point 1.

The major referential support for the use of Ntsr1-Cre mice to target excitatory CN neurons is a preprint (citation #30) produced by the authors of the current study. Since that work has not undergone peer review, the authors should provide stronger rationale and evidence for the use of Ntsr1 VGlut2 mice to target CN neurons. In the preprint, they describe that there are few cells labeled in the fastigial nucleus in the Ntsr1-Cre mice. It would be nice to show this labeling in this paper, since it is important for understanding how much of the nuclear output is being labeled. Their statement from the pre-print seems to weaken their findings, since if they are missing a large population of excitatory outputs from the nuclei, it is hard to conclude that the nucleus is not contributing to social output.

This is a great point raised by multiple reviewers. We have included a new figure, figure 8, to address this point. We have now included a double probe FISH, for *Ntsr1^{Cre}* driven YFP expression and *Vglut2*. We find that the co-expression of YFP and VGlut2 is only present in cerebellar nuclei. We further have quantified the YFP and VGlut2 co-expression. We found a partial overlap and have interpreted the results and modified the discussion accordingly.

Furthermore, the authors state that there are no changes in social behavior in development in Ntsr1 VGlut2 mice, but in fact, do not discuss the apparent delay in the vocalization that they appear to observe in these mice in development, since they show that one measure is significantly different at their oldest developmental age (P11). Again, this seems to weaken the strength of their claims. They could dial back the strength of their claims, but it would also be great if they included further social behaviors, especially in the adult.

We have also included a second new figure, figure 10, that investigates behavior in the adult mice. Our results in the adult mice confirm no change in social behaviors in *Ntsr1^{Cre/+};Vglut2^{fl/fl}* mice.

3. An opportunity to strengthen their findings has been missed by not comparing results from their two mice strains in the adult, since the Ntsr1 VGlut2 mice have not been examined in the adult. If they observed similar results in the Ntsr1 VGlut2 mice as observed in the Atoh VGlut2 adult mice, this would help strengthen their findings, and might in part address the questions raised by the limited Ntsr1 expression.

Thank you for this comment. This concern is closely related to what is discussed in the point above. We have made the suggested comparisons and included the new figures, data, and text descriptions in the revised version of the paper.

4. The author discuss the Atoh VGlut2 knock-out mice as being “rescued” in the adult. A rescue implies that the authors are doing some intervention, but they have not, they are making this claim based on normal development, which (as described in point 1) they have not actually shown. Even if they show that adult granule cell synapses are completely normal, I think that “rescue” is not the appropriate term here, it feels misleading. Please rewrite.

Thank you, we agree with this point. We have rewritten this phrase, removed the said term and instead have called this developmental restoration of function.

Minor Points

1. Typo line 90 “in a social isolation paradigm during.” During what?

Addressed, we deleted “during”.

2. The authors are encouraged to use different stars to show different degrees of significance. Some of their findings are just significant, whereas others are very robust, but they are all shown with 1-star significance, which makes it harder to interpret, and in fact, probably weakens their stronger findings in the eyes of the reader.

We appreciate what the review is conveying. We predetermine our p-value cut-off for significance at $p=0.05$. As this is a predetermined cut-off, we cannot define robustness of the results based on the value of the p-value, especially because the p-value is affected by both the effect size and the sample size. However, we do report exact p-values, effect size, and sample size in our figure legends so that readers can interpret the robustness of our findings. We hope that by providing these different measures the reader will comprehend the key findings and their strengths.

3. The way that the authors represent the 3-chamber test results is not standard, and it would be nice to see more data here, since there could be subtle differences that are not detected with the data presented this way. Since this is a big part of their claim, further social tests would help strengthen these findings.

We changed the representation of the 3-chamber test results to a more standard representation.

4. The authors say that there is labeling in layer 5 in the cortex in the *Ntsr1* mice, and it would be useful to demonstrate if those cells express VGLUT2, and would therefore be affected in these mice.

Good point, and this is one we have discussed above in related to similar comment. To reiterate, we have included a new figure, figure 8, to show overlap between *Ntsr1^{Cre}* and *Vglut2* using a double probe FISH, for *Ntsr1^{Cre}*-driven YFP expression and *Vglut2*. We find that the co-expression of YFP and *Vglut2* is only present in the cerebellar nuclei. We further have quantified the YFP and *Vglut2* co-expression. We found a partial overlap and have interpreted the results and modified the discussion accordingly.

5. Please don't use the same abbreviation for the cochlear nucleus as for the cerebellar nuclei in other figures (CN).

Good catch, thank you. We have changed the abbreviation for cochlear nucleus in Figure 4.

6. Page 4 line 27: Please add a citation for claim that CFs are the only source of VGLUT2 from the *Atoh1* lineage at this age.

Thank you for the suggestion. We have now included references for this claim.

7. The title seems to imply that the authors are examining different CN neurons, rather than comparing CN neurons to other cerebellar neurons (granule cells). Please revise.

Thank you for catching this, we understand and appreciate the concern. We have revised the title (and abstract since we have added some new data) to better represent the data discussed in the manuscript.

Reviewer #3 (Remarks to the Author):

The manuscript by van der Heijden and colleagues uses Cre-flox recombination in mutant mice to delete VGlut2-mediated glutamatergic neurotransmission in Atoh1+ and Ntsr1+ cell lineages. Central to the manuscript is the finding that VGlut2-mediated neurotransmission from Atoh1+ cell lineages disrupts motor reflexes and social vocalizations in juvenile mice compared to the motor only effects of the more restricted deletion of Vglut2 in Ntsr1+ cell lineages. Based on these differences, the authors claim VGlut2 neurotransmission from cerebellar cortical versus (DCN) nuclei neurons differentially controls the acquisition of motor and social behaviors. Unfortunately this claim is not sufficiently supported by the data. The authors need to provide more convincing data supporting the specificity of granule cell versus DCN neuron deletion of VGlut2-mediated neurotransmission in mice. An additional concern are the conclusions made about the comparative neurocircuitry of Atoh1Cre/+;Vglut2fl/fl and control mice based on electrophysiological data with low sampling power.

Results line 167. “substantiating a role for glutamatergic cerebellar neurons in shaping these behaviors.” As it is not possible to separate the effects of ablating VGlut2-mediated neurotransmission in granule cell and DCN lineages from that of extra-cerebellar glutamatergic Atoh+ lineages in Figures 2 and 3 data, the data does not substantiate the role of these cerebellar cell types and this claim and should be removed.

We agree with this comment. We have therefore removed this claim.

Line 173. “mediated by abnormal cortical function” should be edited to “associated with abnormal cortical function”

We have reworded this sentence as suggested.

Supp Figure 1 title typo: “in” should be “is”. Histograms lack +/- SEM bars.

Thank you for catching this typo, we made the change.

Our histograms do not have +/- SEM bars because all data points are shown, which by itself provide a visual indication of the variability of the data. We found that adding the SEM bars to the figures crowded the figures. We appreciate the reviewers concern, although kept the figures as is in order to avoid difficulties in the reader appreciating the finer details of the data points.

Figure 2 and Figures 3D-I, 5B-D, 6D-I. Text should be added to the legend to indicate the larger open circles each represent the mean for the male or female cohorts for the control and mutant strains at each age; a horizontal bar rather than an open circle to denote the mean would be an improvement. It is also not clear what the shaded areas represent. It looks like a way of plotting

the distribution of the data, but the authors should say so if that's the case. It would be a little easier to read if different symbols (as in Fig. 7) or colors were used to represent individual data points for male vs. female. It's true that males and females are in different columns but they're so close together they're hard to distinguish. Clearer annotation of the group means and individual data points to distinguish between groups (males vs females, cells from the same mouse) applies to figure 2 but also figures 3D-I, 5B-D, 6D-I.

We have included better descriptions of the symbols in the figure legends. We also provided a visual depiction of the group mean in the pup behavior data with horizontal lines.

Figure 3 and Figure 6 data. The number of animals and cells sampled (mouse $n = 3$, cells $n = 12-15$) is unusually low. The low sampling power of Figure 3 and 6 data reduces the chance of detecting a true effect in *Atoh1Cre/+; Vglut2fl/fl* mutants when compared to controls. As such the claims that “Vglut2 loss from the *Atoh1* lineage resulted in....no change in simple spike pattern or regularity (Fig. 3E-F)....did not observe any differences in complex spike frequency, pattern, or regularity (Fig. 3H-I)” (lines 183-186) are not sufficiently tested. Similarly, insufficient sampling questions the findings of no difference in simple and complex spike firing in adult *Atoh1Cre/+; Vglut2fl/fl* mutants compared to controls. (Figure 6).

Agreed. We have included recordings from 2 additional mice for each group, reaching a sampling size of $N=5$, and $n=18-24$ cells per group. Increasing the sampling size did not change our findings. We also included the effect size for each comparison in the figure legend.

Figure 3. Text should be added to the legend to indicate CV is Purkinje cell spike pattern and CV2 Purkinje cell spike regularity

We have included this text in the legends.

Figure 4 – Supplement 1. No mention is made of the arrow annotations in the legend.

Thank you for catching this. We have revised legend and included a description.

Figure 5. Restricted ablation of VGLUT2-mediated neurotransmission in DCN neurons is predicated on *Ntsr1-Cre* recombinase-mediated deletion. However, other than a reference to Allen Brain Atlas data, no data on *Ntsr1* expression in the DCN is provided and it is not clear to what extent an *Ntsr1-Cre; VGLUT2fl/fl* background would deplete Vglut2 in the DCN. Indeed no data is included demonstrating how effective and expansive this approach is in deleting Vglut2 in the DCN. At a minimum, the authors should provide comparative VGLUT2 expression data in the DCN *Ntsr1-Cre; VGLUT2fl/fl* mice and controls. Ideally this would be supported by the inclusion of detailed *Ntsr1-Cre* fate-mapping in the DCN to identify the targeted nuclei.

Thank you, this is a great idea. We have now included a Cre fate-mapping experiment in the revised version of the paper. The data from this experiment are presented in the new figure 8.

Figure 7A-C. Tremor effects are mild. How does the magnitude of the decrease in tremor activity in *Atoh1Cre/+; VGLUT2fl/fl* mice compare to the change observed with deleting GABAergic neurotransmission from Purkinje cells?

Thank you for raising this point. We have discussed this further and have now included in the text a description indicating that the reduction in tremor in the *Atoh1* mutants is similar to that observed in mice without GABAergic neurotransmission from Purkinje cells.

Figure 7. For consistency, the authors should annotate the means for female and males.

We plotted all individual data points for male and female mice. We did not observe statistical differences in sex and have reported all the data as one group. Therefore, we did not annotate the means for females and males separately. We have included these statements in the text.

Results text (lines 242-247) discussing the possibility social deficits recorded in the early postnatal *Atoh1*Cre/+; *Vglut2*fl/fl mice are caused by abnormal function within the cerebellar cortex are speculative and better placed in the discussion section.

Thank you, we agree. We have removed these statements from the results section.

The closing results section statement (lines 296-298) is not sufficiently supported by the data – see Fig. 5 concerns with the lack of data supporting *VGLuT2* deletion in *Ntsr1*-Cre; *VGLuT2*fl/fl mice: “Together, our findings indicate that altering fast neurotransmission from glutamatergic cerebellar nuclei neurons in the developing cerebellar circuit later obstructs the proper execution of motor functions but does not impair social behaviors in adult mice.”

Agreed. We have made this statement more specific to only reflect the data regarding the *Atoh1* lineage neurons.

Discussion. Similar to the results sections, the statements “genetic elimination of neurotransmission from glutamatergic nuclei neurons did not impair the acquisition of social vocalizations in early postnatal mice” and “We conclude that intact *VGLuT2*-mediated neurotransmission from glutamatergic nuclei neurons in postnatal mice is essential for the acquisition and coordination of movements, but it is not required for the acquisition and maintenance of social behaviors that require intact cerebellar function” are not sufficiently supported by the data.

We have reworded this paragraph to better reflect our data and new findings.

Are there weight differences between *Atoh1*Cre/+; *Vglut2*fl/fl (or *Ntsr1*-Cre; *Vglut2*fl/fl) and control pups at the ages tested for motor and vocal activity during development? The inclusion of weight data would assuage a concern differences are confounded by developmental delay in the mutants.

We did not observe weight differences between conditional knockout and control mice at multiple different time points. We have now included these data and statements in the methods.

Statistical analysis. Are all data equally balanced for sex? If so, please state. If not and the data is separated by sex in figures, the n of each sex per group should be included in the legends.

This is a good point; we have now included the n for each sex, per group, in the figure legends.

REVIEWERS' COMMENTS

Reviewer #1 (Remarks to the Author):

It was a real pleasure to read the revised version of this manuscript. Again the quality of the histological control is remarkable. The substantial changes made by the authors greatly improved the manuscript and made their findings even more convincing.

Reviewer #2 (Remarks to the Author):

The authors have addressed my concerns and I think that the revised manuscript is significantly strengthened.

Reviewer #3 (Remarks to the Author):

I thank the authors for addressing each of my comments and commend them on the quality and scope of the additional data, particularly the inclusion of *Ntsr1^{Cre}* genetic-inducible fate-map (GIFM) images. Although not clear in the Figure 8 data, public database images for this *Ntsr1^{Cre}* line mark neuronal lineages in the medulla and scattered fiber tracts in the white matter and cortex of the cerebellum. This raises the possibility of alterations in network activity in the cerebellar cortex of *Ntsr1^{Cre}*; *Vglut2^{fl/fl}* mice that ought to be addressed by the authors.

Line 209 typo: "ln" should be "in"

REVIEWER COMMENTS

We would like to thank all three reviewers for providing excellent suggestions that have enabled us to strengthen our manuscript and enhance the impact of the findings. We have addressed the final set of revisions by adding some figure panels and changing the text.

Below are our explanations for how we have altered the manuscript in this revised version. The Reviewer's comments are written in black, and our responses are written in blue.

Reviewer #3 (Remarks to the Author):

I thank the authors for addressing each of my comments and commend them on the quality and scope of the additional data, particularly the inclusion of *Ntsr1Cre* genetic-inducible fate-map (GIFM) images.

Thank you for commending us on the quality and scope of the data.

Although not clear in the Figure 8 data, public database images for this *Ntsr1Cre* line mark neuronal lineages in the medulla and scattered fiber tracts in the white matter and cortex of the cerebellum. This raises the possibility of alterations in network activity in the cerebellar cortex of *Ntsr1Cre;Vglut2fl/fl* mice that ought to be addressed by the authors.

We have not found any *YFP* (driven by *Ntsr1-Cre*) positive neurons that also express *Vglut2* outside the cerebellar nuclei. We have provided additional insets in the striatum, midbrain, and medulla to show that the sporadic *YFP*-positive neurons are not expressing *Vglut2* (Figure 8a). We have additionally underscored these findings in the text:

“In agreement with previous studies, we found *Ntsr1^{Cre}*-driven *YFP* expression in the cerebellar nuclei neurons (Fig. 8a inset iii). We also found *YFP* expression in layer 6 neurons of the cerebral cortex (Fig. 8a inset i), and sparse labeling in the striatum (Fig. 8a inset iv), midbrain, (Fig. 8a inset v), and medulla (Fig. 8a inset vi). While there was some *Vglut2* expression in similar areas in the midbrain and medulla, we did not observe any co-expression of *YFP* and *Vglut2* in the same neurons in any brain region other than the cerebellar nuclei (Fig. 8a inset iii).”

Because we do not see overlap between *Ntsr1-Cre* driven *YFP* expression and *Vglut2* we believe that the source of the fiber tracts observed in the cerebellar cortex are the cerebellar nuclei. Indeed, in our intersectional lineage tracing, we see many projections from the cerebellar nuclei to the cerebellar cortex (supplemental Figure 7). We have also discussed this in the text:

“We also see some tdTomato+ projections traveling to the molecular layer within the cerebellar cortex (Supplemental Figure 7b), which may explain the presence of fibers tracks in the cerebellar cortex observed in the initial description of this mouse line.”

Line 209 typo: “In” should be “in”

Thank you for catching this typo. We have addressed it.